# Visualizing subcellular changes in the NAD(H) pool size versus redox state using fluorescence lifetime imaging microscopy of NADH

Angela Song [1,2], Nicole Zhao[1], Diana C. Hilpert[3], Caroline Perry [4], Joseph A. Baur[4], Douglas C. Wallace [1,5] ✉ & Patrick M. Schaefer [1] ✉

NADH autofluorescence imaging is a promising approach for visualizing energy metabolism at the single-cell level. However, it is sensitive to the redox ratio and the total NAD(H) amount, which can change independently from each other, for example with aging. Here, we evaluate the potential of fluorescence lifetime imaging microscopy (FLIM) of NADH to differentiate between these modalities. We perform targeted modifications of the NAD(H) pool size and ratio in cells and mice and assess the impact on NADH FLIM. We show that NADH FLIM is sensitive to NAD(H) pool size, mimicking the effect of redox alterations. However, individual components of the fluorescence lifetime are differently impacted by redox versus pool size changes, allowing us to distinguish both modalities using only FLIM. Our results emphasize NADH FLIM's potential for evaluating cellular metabolism and relative NAD(H) levels with high spatial resolution, providing a crucial tool for our understanding of aging and metabolism.

Alterations in cellular energy metabolism and redox state are hallmarks of a variety of disorders such as cancer, diabetes, and neurodegenerative diseases[1,2]. In many such diseases, metabolic alterations show tissue, cell, and even mitochondrial specificity[3,4]. Consequently, new approaches to evaluate metabolism with high spatial resolution are needed.

One promising method relies on the autofluorescence of the electron carrier nicotinamide adenine dinucleotide (NADH)[5]. Because glycolysis and the citric acid cycle reduce $NAD^+$ to NADH, while the electron transport system re-oxidizes NADH back to $NAD^+$, the ratio of $NAD^+$ to NADH can reveal shifts in cellular energy metabolism that favor mitochondrial respiration or glycolysis. Importantly, NADH is autofluorescent while $NAD^+$ is not, so higher NADH fluorescence intensity is associated with a more reduced redox state and vice versa[6]. Accordingly, it has been shown that blocking mitochondrial respiration results in increased NADH intensity[7] while blocking glycolysis results in lower NADH intensity[8]. As such, NADH imaging enables an estimation of cellular energy metabolism

and mitochondrial function microscopically, without external markers or reporters, which renders it an ideal candidate for in-vivo applications.

To avoid drawbacks of intensity imaging in tissues, such as differences in absorption or light scattering, NADH fluorescence lifetime imaging microscopy (NADH FLIM) can be used. NADH FLIM relies on the differences in the fluorescence lifetimes between free NADH (around 400 ps) and protein-bound NADH (around 2500 ps)[9]. Since the protein-bound NADH pool remains relatively stable in comparison to the free NADH pool, a larger protein-bound to free NADH ratio, or a longer mean NADH lifetime, indicates less NADH relative to $NAD^+$ and a more oxidized redox state. Multiple studies have shown that blocking mitochondrial respiration, which is expected to increase the free NADH pool, results in a shorter NADH lifetime[10–12], establishing the use of NADH fluorescence lifetime as a promising surrogate marker for cellular energy metabolism.

However, $NAD^+$ and NADH are involved in a multitude of redox reactions in the cell, meaning a variety of factors independent of energy

---

[1]Center for Mitochondrial and Epigenomic Medicine, Department of Pathology and Laboratory Medicine, Children's Hospital of Philadelphia, Philadelphia, PA, USA. [2]Department of Bioengineering, University of Pennsylvania, Philadelphia, PA, USA. [3]Department of Biological Sciences, Dartmouth College, Hanover, NH, USA. [4]Department of Physiology and Institute for Diabetes, Obesity, and Metabolism, Perelman School of Medicine, University of Pennsylvania, Philadelphia, PA, USA. [5]Department of Pediatrics, Perelman School of Medicine, University of Pennsylvania, Philadelphia, PA, USA. ✉e-mail: wallaced1@chop.edu; schaeferpm@chop.edu; pschaefer@cell.com

metabolism influence the detected NAD(H) redox state[13]. These include the pH, which is especially variable in the mitochondrial matrix[10], and protein composition, which impacts the lifetime of the protein-bound NADH[14]. Additionally, NADPH autofluorescence cannot be spectrally distinguished from NADH fluorescence[15]. As a result, these factors must be taken into account when utilizing NADH imaging to evaluate cellular metabolism.

NAD(H) pool size, defined as the total amount of $NAD^+$ and NADH combined, is another variable that must be considered. Importantly, NAD(H) pool size can be altered in many diseases, such as colon cancer[16] and mitochondrial myopathy[17]. Likewise, aging is associated with decreased $NAD^+$ levels and a decreased NAD(H) pool due to dysfunction in $NAD^+$ biosynthesis[18,19]. Accordingly, supplementation with $NAD^+$ precursors, which not only affect $NAD^+$ levels but to a lesser extend also NADH, has recently gained much attention[20]. Thus, a non-invasive method to assess NAD(H) pool sizes in addition to redox state and metabolism would be extremely valuable.

However, increases in NAD(H) pool size cannot be distinguished from a more reduced redox state in NADH intensity imaging, since both cause an increase in fluorescence intensity. Thus far, though, the impact of changes in the NAD(H) pool size on NADH lifetime from FLIM have not been investigated.

Here, we hypothesize that combining NADH intensity and lifetime information using NADH FLIM can allow us to simultaneously evaluate alterations in the NAD(H) pool and redox state, providing an extremely valuable tool, for example, to study aging. To test this, we perform targeted modifications of NAD(H) pool size in cells and mice. We show that pool size is inversely correlated with the mean NADH lifetime, and subsequently establish different approaches to differentiate between pool size-induced and respiration-induced changes in NADH autofluorescence.

## Results

### Changes in NAD(H) pool size impact NADH FLIM

To determine if NADH FLIM is sensitive to changes in NAD(H) pool size, we first modified pool size by treating HEK293 cells with either nicotinamide riboside (NR), which feeds into the $NAD^+$ salvage pathway, to increase pool size, or FK866, an $NAD^+$ salvage pathway inhibitor, to decrease pool size. Interestingly, treatment with NR caused a significant decrease in the mean NADH lifetime (τmean) in the mitochondria, but not the nucleus or cytoplasm (Fig. 1a). Conversely, treatment with FK866 significantly increased τmean in the mitochondria, nucleus, and cytoplasm (Fig. 1b). These results suggest that the NADH lifetime is sensitive to changes in the NAD(H) pool size.

We next cross-validated these findings in 143B osteosarcoma cells to ensure that the effects of NAD(H) pool size on NADH FLIM are not cell type specific. Similar to HEK293 cells, treatment with NR decreased mean NADH lifetime relative to control cells, and treatment with FK866 increased mean lifetime (Fig. S1a, b). Cytoplasmic NADH was not analyzed separately since it equilibrates with the nucleus through the nuclear pores[21]. This was confirmed by correlating nuclear and cytoplasmic mean NADH lifetimes across our treatment (Fig. S2).

To validate that the observed changes in mitochondrial, nuclear, and cytoplasmic τmean upon NR and FK866 treatments were in fact due to alterations in NAD(H) pool size and not due to changes in the redox ratio, we first assessed $NAD^+$ and NADH levels biochemically. As expected, titrations of both NR and FK866 in HEK293 cells revealed dose dependent alterations of NAD(H) pool size (Fig. 1c). Importantly, the biochemically quantified $NAD^+$/NADH ratio was not significantly different for any condition (Fig. 1d), suggesting that neither treatment strongly altered cellular redox state. Unfortunately, at 5 nM FK866, we could not assess the $NAD^+$/NADH ratio reliably from the biochemical assays since NAD(H) levels were too low. Thus, to avoid bias, we performed NADH FLIM of HEK293 cells treated with varying concentrations of FK866 and NR, which showed that FK866 increased τmean and NR decreased τmean in a concentration dependent manner (Fig. S3a, b).

To further verify that neither treatment significantly impacted NADH lifetimes by modifying energy metabolism, we performed Oroboros high resolution respirometry on cells treated with 300 μM NR and 5 nM FK866. We revealed that routine respiration, which mirrors the respiratory state during FLIM measurements, was not altered (Fig. 1e). Likewise, electron transport system capacity (ETS capacity), leak respiration, and the respiratory control ratios were not significantly altered, and neither was doubling time (Fig. S3c–h). Additionally, we measured lactate secretion rate. Similar to respiration, treated cells showed no significant difference in lactate secretion compared to controls (Fig. 1f), suggesting no change in glycolysis. Finally, to additionally confirm neither NR nor FK866 impact metabolism, we performed Seahorse metabolic flux assays on 143B and HEK293 cells treated with NR and FK866 to determine basal respiration. Oxygen consumption rate (OCR) was not effected by either treatment in 143B cells (Fig. S4a), but 5 nM FK866 treatment significantly lowered OCR in HEK293 cells (Fig. S4b, c). This was only observed at the highest FK866 concentration, and is not totally surprising given that that 5 nM FK866, NAD(H) levels are nearly undetectable (Fig. 1c). The OCR/ECAR ratio was not impacted by either treatment in 143B cells (Fig. S4d), but was decreased in HEK293 cells treated with FK866 (Fig. S4e, f). Thus, FK866 treatment shifts metabolism in HEK293 cells towards glycolysis. However, this cannot explain the lifetime changes we observe, because a decreased OCR/ECAR ratio suggests a more reduced redox ratio, which would shorten the NADH lifetime. Instead, we observe significantly increased lifetime upon FK866 treatments, recapitulating that the altered lifetimes upon NR and FK866 treatment are not caused by differences in cellular energy metabolism.

Since NADPH autofluorescence cannot be distinguished from NADH fluorescence spectrally, we also quantified NADP(H) levels biochemically. As expected, NADP(H) levels follow similar trends as NAD(H) levels upon NR and FK866 treatments (Fig. S5a,b). Comparing the relative amounts of NADPH versus NADH revealed that NR treatment did not change the ratio of NADPH to NADH, but FK866 treatment resulted in higher NADPH/NADH ratio (Fig. S5c). To verify our biochemical quantifications, we also performed analysis developed by Blacker et al. to determine the NADPH/NADH ratio in treated and untreated cells using lifetime components from FLIM[15] (Fig. S5d, e). This analysis showed close correlations between the optically quantified NADPH/NADH ratio in both mitochondria and the nucleus and whole cell biochemical quantifications of NADPH/NADH, suggesting no significant differential effects on the subcellular level (Fig. S5f, g). To confirm this, we isolated cytosolic and mitochondrial fractions from NR and FK866 treated cells, and biochemically quantified NADPH/NADH in each subcellular fraction and in whole cell samples collected prior to fractionation (Fig. S5h). In accordance with prior studies, we found that NADPH/NADH is higher in cytosol than in mitochondria[22]. Crucially though, treatment with both FK866 and NR has a similar effect on NADPH/NADH in mitochondria and in cytosol compared to whole cells, reiterating that neither NR or FK866 causes differential changes to the NADPH/NADH ratio across cellular compartments. Thus, their impact on subcellular NADH FLIM results is also minimal.

Given that NADPH has been reported to have a longer protein-bound lifetime compared to NADH[15], the increased fraction of NADPH might, in part, explain the longer mean NAD(P)H lifetime we observed upon FK866 treatment. However, given that FK866 affected nuclear and mitochondrial τmean similarly, although the NADPH to NADH ratio is known to typically be higher in nuclei compared to mitochondria, it is highly unlikely that the entire FK866 effect is mediated by higher NADPH/NADH. Similarly, the NAD(P)H lifetime alterations upon NR treatment cannot be explained by changes in NADPH.

Taken together, our results demonstrate, that in contrast to expectations, changes in the NAD(H) pool size do not only affect the NADH autofluorescence intensity but also the mean NADH lifetime, resulting in an inverse relationship between pool size and mean NADH lifetime.

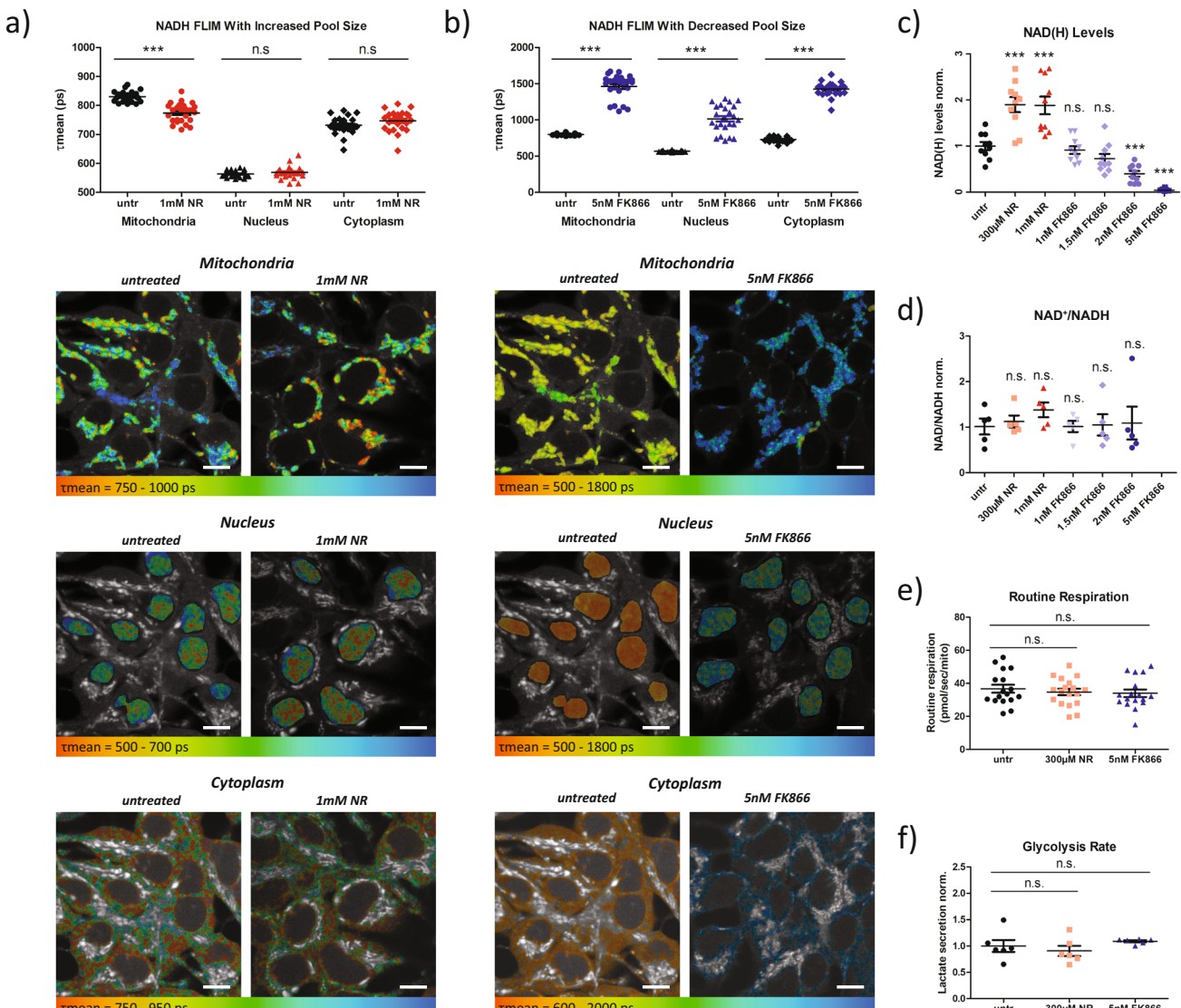

**Fig. 1 | NADH FLIM is sensitive to changes in the NAD(H) pool size. a** Mean NADH lifetime (τmean) of mitochondria, nuclei, and cytoplasm in untreated and nicotinamide riboside (NR) treated HEK293 cells (*n* = 25 for both) with exemplary images encoding τmean in false-colors (red = shorter, blue = longer). Scale bars represent 10 microns. **b** Mean NADH lifetime (τmean) of mitochondria, nuclei, and cytoplasm in untreated and FK866 treated HEK293 cells (*n* = 25 for both) with exemplary images encoding τmean in false-colors (red = shorter, blue = longer). Scale bars represent 10 microns. NAD(H) pool size (**c**) and ratio (**d**) quantified biochemically upon different concentrations of NR and FK866 (*n* = 10 for all groups

in (**c**), *n* = 5 for all groups in (**d**)). **e** Mitochondrial respiration (Routine respiration) of intact untreated HEK293 cells (*n* = 17), NR treated HEK293 cells (*n* = 18), and FK866 treated HEK293 cells (*n* = 17). **f** Lactate secretion rate as an estimate of anaerobic glycolysis (*n* = 6 for all groups). Each data point is independent. The bars indicate the mean and standard error. Significances were calculated using t tests between selected groups in (**a–d**) and using ANOVA and Dunn's post hoc test between selected groups in (**e, f**). For (**c**) and (**d**), significances are calculated against the untreated (untr) group. Significances are indicated as n.s. for not significant and *** for *p* < 0.001.

## NADH FLIM allows visualization of NADH pool size differences on a single cell level

To demonstrate that NADH FLIM can be used to evaluate NAD(H) pool size or redox changes even on a single cell level, we expressed mitochondrially targeted *Lactobacillus brevis* H₂O-forming NADH oxidase (mt*Lb*NOX)[23] in cells. We created a bicistronic plasmid which expressed the fluorescent protein CayenneRFP and mt*Lb*NOX to similar levels. This allowed us to estimate mt*Lb*NOX expression by measuring fluorescence intensity of CayenneRFP in each cell. Stable cell lines were established without clonal selection, resulting in heterogeneous expression of mt*Lb*NOX across individual cells.

First, we biochemically assessed the effect of mt*Lb*NOX expression on both NAD⁺ and NADH levels in 143B osteosarcoma cells (Fig. 2a–c). Contrary to Titov et al.'s findings in their original paper characterizing mt*Lb*NOX expressing cells[23], we found a significant decrease in both NAD⁺

and NADH levels (Fig. 2d, e). Subsequent testing with an alternate biochemical quantification kit confirmed these findings and demonstrated no significant difference in NAD⁺/NADH ratio between control and mt*Lb*NOX expressing cells (Fig. S6a–c), possibly because the cell naturally equilibrates the redox state even when only NADH is scavenged by mt*Lb*NOX. To test this hypothesis, we quantified NAD⁺ and NADH one week and two weeks (data in Fig. S6a–c) after starting selection pressure with G418 in our mt*Lb*NOX cells (Fig. S6d–f). At the earlier 1-week time point, the redox ratio is more reduced in mt*Lb*NOX expressing cells, which more closely matches Titov et al.'s results[23]. However, NAD⁺ levels were already significantly decreased, suggesting that the cells had begun to equilibrate their redox ratio. These results demonstrate that the time after transfection and the respective cellular response to *Lb*NOX expression play a significant role in the effect of *Lb*NOX on NAD(H) pool size and redox ratio, and suggest that our mt*Lb*NOX overexpressing cells can be used as an

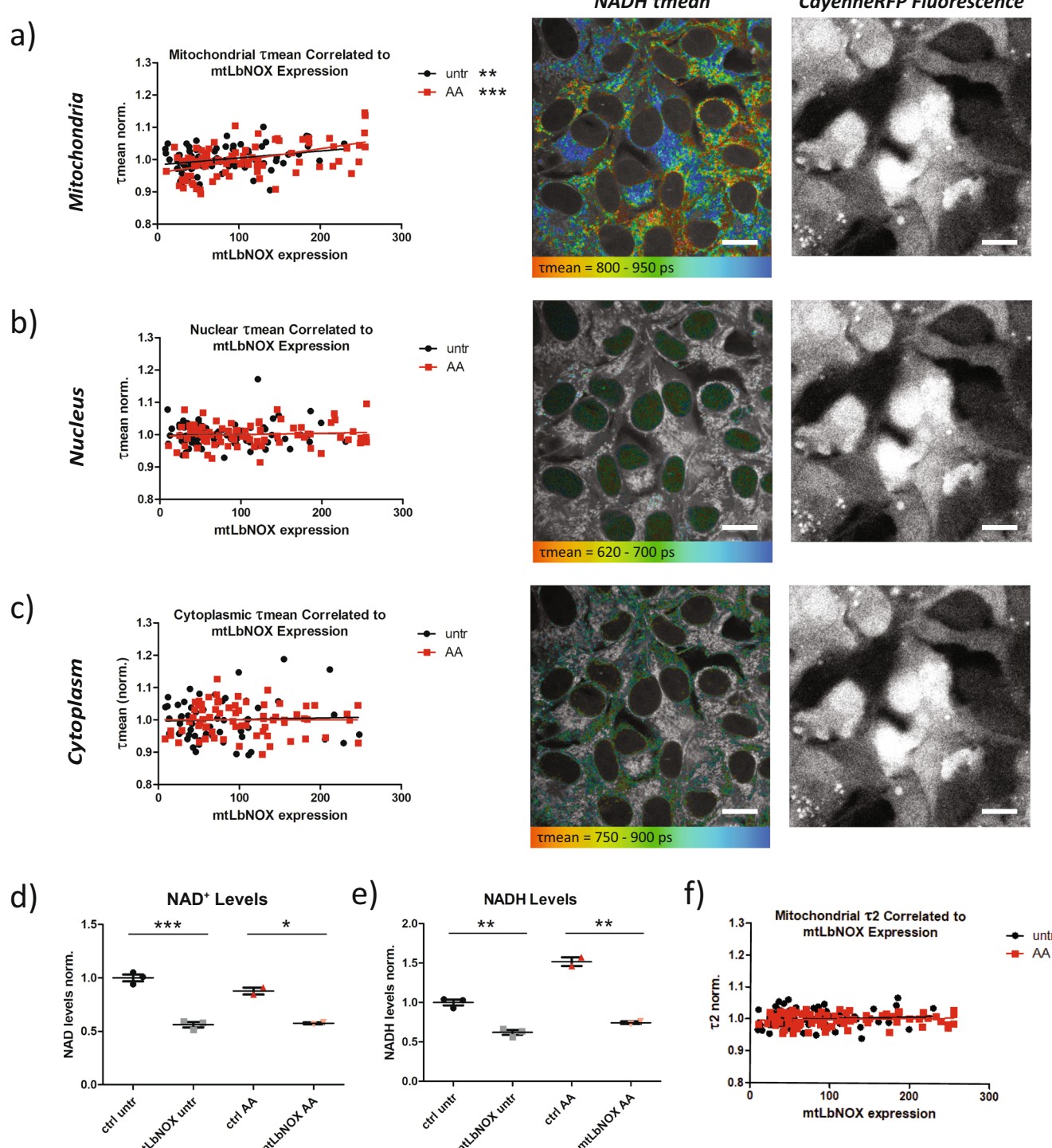

**Fig. 2 | mt*Lb*NOX supports the effect of NAD(H) pool size on NADH FLIM.**
**a** Mean NADH lifetime (τmean) in mitochondria of antimycin A (AA) treated 143B cells normalized to average mitochondrial τmean in all AA treated cells (red), and in mitochondria of untreated 143B cells normalized to average mitochondrial τmean in all untreated cells (black), correlated to mt*Lb*NOX expression ($n = 86$ for untr, $n = 80$ for AA). **b** Mean NADH lifetime (τmean) in nuclei of AA treated 143B cells normalized to average nuclear τmean in all AA treated cells (red), and in nuclei of untreated 143B cells normalized to average nuclear τmean in all untreated cells (black), correlated to mt*Lb*NOX expression ($n = 86$ for untr, $n = 80$ for AA). **c** Mean NADH lifetime (τmean) in cytoplasm of AA treated 143B cells normalized to average cytoplasmic τmean in all AA treated cells (red), and in cytoplasm of untreated 143B cells normalized to average cytoplasmic τmean in all untreated cells (black), correlated to mt*Lb*NOX expression ($n = 57$ for untr, $n = 72$ for AA). mt*Lb*NOX expression was quantified by CayenneRFP fluorescence intensity.

Representative FLIM images encode τmean in false-colors (red = shorter lifetime, blue = longer lifetime) and corresponding fluorescence intensity images show CayenneRFP fluorescence (used as a direct measure of mt*Lb*NOX expression levels). NAD⁺ (**d**) and NADH (**e**) levels quantified biochemically with the NAD/NADH Glo™ Assay (Promega) upon expression of mt*Lb*NOX and AA treatment ($n = 3$ for untr, $n = 2$ for AA). **f** Protein-bound NADH lifetime (τ2) in mitochondria of AA treated 143B cells normalized to average mitochondrial τ2 in all AA treated cells (red), and in mitochondria of untreated 143B cells normalized to average mitochondrial τ2 in all untreated cells (black), correlated to mt*Lb*NOX expression ($n = 86$ for untr, $n = 80$ for AA). Each data point is independent. Scale bars represent 20 microns. Correlations and significances for (**a**), (**b**), (**c**), and (**f**) were calculated using linear regression. The bars for (**d**) and (**e**) indicate the mean and standard error, and significances were calculated using $t$ tests between selected groups and are indicated as * for $p < 0.05$, ** for $p < 0.01$, and *** for $p < 0.001$.

appropriate tool to study the effect of a reduced NAD(H) pool size but stable redox ratio on NADH FLIM.

Because mt*Lb*NOX expression varied between cells, the mean NADH lifetime in the nucleus and mitochondria was quantified for each individual cell. Simultaneously, CayenneRFP fluorescence intensity was measured as a surrogate marker of mt*Lb*NOX expression of each cell. Next, we correlated the mean NADH lifetime to mt*Lb*NOX expression, revealing a positive correlation for mitochondrial τmean (Fig. 2a). No correlation was seen for nuclear or cytoplasmic τmean (Fig. 2b, c), consistent with the mitochondrial localization of mt*Lb*NOX. This can be seen in side-by-side comparisons of CayenneRFP fluorescence intensity images (Fig. 2a–c, right images) and false-color coded images of the mean NADH lifetime (Fig. 2a–c, left images). Cells with brighter CayenneRFP fluorescence clearly showed longer mitochondrial NADH lifetimes (top left image, more blue) but no alteration in the nuclear or cytoplasmic NADH lifetime (middle and bottom left images). The lack of alterations in nuclear and cytoplasmic NADH lifetime additionally confirmed that our expression marker, Cayenne RFP, was not impacting NADH lifetime measurements.

Since mt*Lb*NOX itself consumes oxygen[23], standard respirometry could not be performed to rule out changes in cellular energy metabolism. To validate that the observed variation in NADH lifetimes between high and low mt*Lb*NOX expressing cells was not due to differences in metabolism, cells were treated with the respiratory chain inhibitor antimycin A (AA)[10]. By eliminating respiration in all cells independent of their mt*Lb*NOX expression levels, any remaining correlation between mt*Lb*NOX and NADH lifetime could not be due to differences in mitochondrial respiration. Indeed, we still observed a significant correlation between mt*Lb*NOX expression and mitochondrial NADH lifetime (Fig. 2a, red line) but not in the nucleus or cytoplasm (Fig. 2b, c, red line). Furthermore, we detected no differences in the slopes of the correlations before and after antimycin A treatment (Fig. 2a–c, red and black lines), indicating that the effect of mt*Lb*NOX on the mitochondrial lifetime is not mediated by mt*Lb*NOX altering mitochondrial respiration and redox state.

Next, to rule out the possibility that binding of NADH to mt*Lb*NOX was impacting the protein-bound NADH fluorescence lifetime, we plotted τ2, representative of the protein bound NADH lifetime, against the mt*Lb*NOX expression (Fig. 2f). Unlike τmean, τ2 showed no significant correlation with mt*Lb*NOX expression, suggesting that mt*Lb*NOX expression does not significantly impact the protein-bound NADH lifetime.

To sum up, we demonstrated that NADH FLIM can visualize pool size differences on a single cell level between adjacent cells in a heterogeneous population, and further validated that pool size and mean NADH lifetime are inversely correlated.

## NADH FLIM can detect changes in NADH pool size in tissues

NR has been proposed as treatment for a large number of pathological conditions and age related diseases by increasing NAD+ levels or NAD(H) pool size[20]. To assess whether pharmacological modifications to the NAD(H) pool size in-vivo can be visualized within tissues, mice were injected with 500 mg/kg NR or 0.9% NaCl. As the NAD+ salvage pathway is most prominent in the liver, liver tissue was dissected and imaged three hours after mice were injected with treatments.

When performing NADH FLIM of liver tissue, we observed bright spots of autofluorescence in our NADH channel (emission 430–490 nm) that demonstrated a short lifetime component of around 1000 ps (Fig. S7a, right images). This signal was present in all mice, regardless of genotype or treatment group, and interestingly, seemed to increase in frequency but not intensity with age (Fig. S7a, lower vs. upper images). As 1000 ps is not a typical NADH fluorescence lifetime component, we hypothesized that these bright spots were due to non-NADH autofluorescent molecules. This was confirmed by the absence of a response of this signal to antimycin A, which blocks respiration and thereby reduces the NADH lifetime[10]. In the phasor plot, a graphical representation of fluorescence lifetime characteristics, antimycin A treatment caused a shift in the population of pixels that represented true NADH signal in the phasor plot, but no shift in the unknown population, confirming that the bright spots did not represent NADH (Fig. S7b). Thus, in subsequent analyses, the region on the phasor plot which corresponded to NADH fluorescence was selected to exclude non-NADH fluorescence (Fig. S7c). Given the fluorescence characteristics of this unknown signal and its increase with age, we hypothesize that it might represent lipofuscin.

As expected, liver samples from mice treated with NR showed a shorter mean NADH lifetime (Fig. 3a). Interestingly, we observed strong regional differences in the mean NADH lifetime within individual liver pieces. This might be due to metabolic differences within liver lobules, suggesting NADH FLIM can be a promising technique to reveal these metabolic differences in future studies. To verify the effect of NR on liver nucleotide levels, we quantified NAD+ and NADH. Indeed, NR increased liver NAD+ and NADH levels to a similar extent (Fig. 3b), demonstrating an increased NAD(H) pool size but no redox changes.

Overall, these data demonstrate that changes in NAD(H) pool size directly result in inverse changes in the mean NADH lifetime across

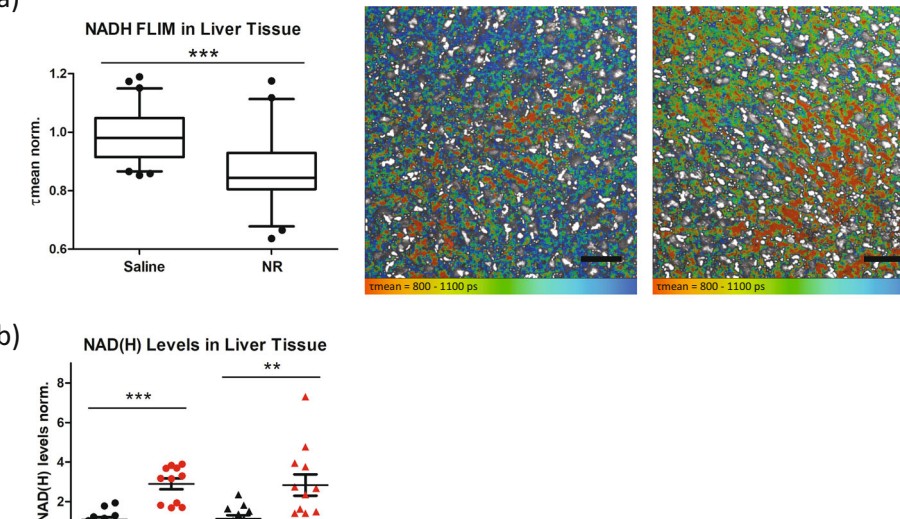

**Fig. 3 | NADH pool size impacts FLIM imaging of mouse liver tissues. a** Mean NADH lifetime (τmean) of liver tissue for mice injected with saline (*n* = 60) or 500 mg/kg NR (*n* = 58). Representative FLIM images encode τmean in false-colors (red = shorter lifetime, blue = longer lifetime). Scale bars represent 20 microns. **b** NAD(H) levels quantified biochemically on tissues from mice treated with saline and NR (*n* = 12 for saline NADH, saline NAD+, and NR NAD+, *n* = 11 for NR NADH). Box in (**a**) shows mean and quartiles while whiskers show 95% confidence interval and data points represent outliers. Bars in (**b**) indicate the mean and standard error. Each data point is independent. Significances were calculated using *t* tests between selected groups and are indicated as ** for *p* < 0.01 and *** for *p* < 0.001.

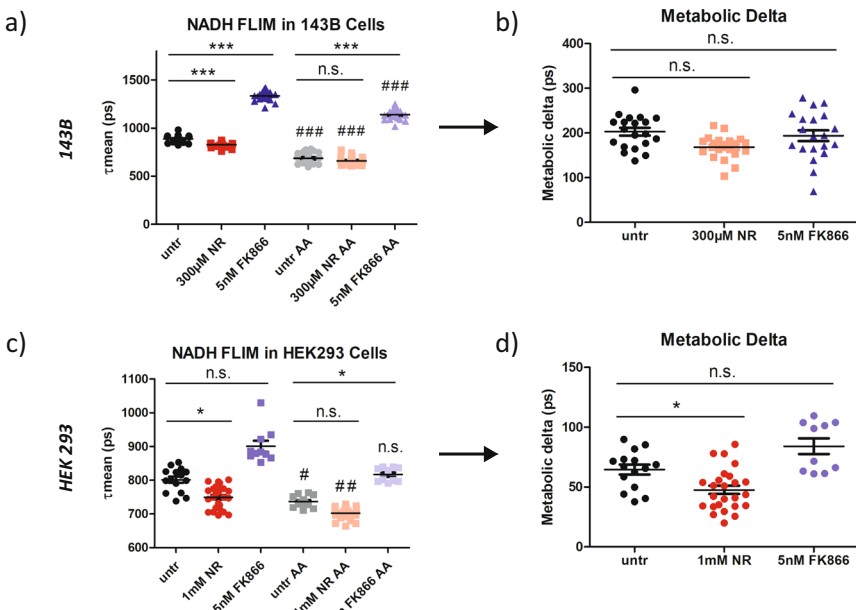

**Fig. 4 | Metabolic delta corrects for the impact of NAD(H) pool size in mito-chondria. a** Mean NADH lifetime (τmean) of mitochondria in untreated, NR treated, and FK866 treated 143B cells before and after antimycin A (AA) treatment (1 μM) (*n* = 20 for all groups). **b** Metabolic delta of untreated, NR, and FK866 treated 143B cells calculated by subtracting individual points for mean lifetime after AA treatment from average mean lifetime before AA treatment for each treatment group (n = 20 for all groups). **c** Mean NADH lifetime (τmean) of mitochondria in untreated, NR treated, and FK866 treated HEK293 cells before and after antimycin A (AA) treatment (1 μM) (*n* = 15 for untr, *n* = 25 for 1 mM NR, *n* = 10 for 5 nM

FK866). **d** Metabolic delta of untreated, NR, and FK866 treated HEK293 cells cal-culated by subtracting individual points for mean lifetime after AA treatment from average mean lifetime before AA treatment for each treatment group (*n* = 15 for untr, *n* = 25 for 1 mM NR, *n* = 10 for 5 nM FK866). Each data point is independent. The bars indicate the mean and standard error. Significances were calculated using ANOVA with Dunn's post hoc test between selected groups for multiple compar-isons and are indicated as n.s. for not significant, * for *p* < 0.05, and *** for *p* < 0.001.

different cell lines as well as in tissues. On a positive note, this allows assessment of the NAD(H) pool size optically with high spatial resolution using NADH FLIM. On the other hand, it complicates the interpretation of the mean NADH lifetime as a marker for either energy metabolism, NAD(H) redox state, or NAD(H) pool size alone. Thus, approaches must be established to distinguish the underlying biological modalities that can affect the mean NADH lifetime.

## Inhibition of respiration enables separation of respiration induced and non-respiration induced NADH FLIM changes

Thus far, we have shown that NAD(H) pool size impacts NADH FLIM independent of respiration. However, when NADH FLIM is being used as an indirect marker of cellular energy metabolism, simultaneous alterations in the NAD(H) pool size can confound those measurements. Thus, in order to evaluate metabolic changes, it is essential to isolate respiration-induced changes in NADH FLIM from changes induced by other factors. To accomplish this, cells were treated with antimycin A, a complex 3 inhibitor, immediately before the FLIM measurement. This eliminates any changes in the mitochondrial redox ratio due to respiration but does not impact many other factors, like total cellular NAD(H) pool size, which may confound the FLIM measurements when assessing cellular metabolism. Thus, if a dif-ference between treatment and control persists after AA treatment, it is not due to respiration.

We applied this concept to NR and FK866 treated 143B and HEK293 cells, of which we know NAD(H) pool size is altered (Fig. 1c), while respiration remains largely unchanged (Fig. 1e). As demonstrated pre-viously (Fig. 1, Fig. S1, Fig. S3), NR resulted in a shorter and FK866 in a longer mitochondrial mean NADH lifetime. These differences persisted upon treatment with AA (Fig. 4a, c), demonstrating again that the differ-ences in NADH lifetime between NR, FK866, and untreated cells cannot be due to differences in respiration, since mitochondrial respiration was zero across all conditions following AA.

By calculating the difference between the mean NADH lifetime with and without respiration-inhibition, which we term metabolic delta, the impact of respiration on NADH FLIM can be isolated in a single parameter. Similar metabolic delta values between populations suggest similar levels of respiration, regardless of the absolute mean lifetime values. Performing metabolic delta calculations on the NR and FK866 treated 143B cells revealed similar metabolic delta values for all 3 treatments (Fig. 4b), reiterating that the changes we see in NADH FLIM after NR and FK866 treatment are not due to metabolic changes. However, metabolic delta of NR treated HEK293 cells was slightly lower than untreated controls (Fig. 4d). This difference was significant, despite no change in mitochondrial respiration (Fig. 1e), and the same trend was observed in 143B cells, although it did not reach significance.

To further understand this limitation, we plotted the total cellular NAD(H) levels after NR and FK866 treatment against the respective average τmeans (Fig. S8a). This correlation demonstrates that the relationship between NAD(H) pool size and mean NADH lifetime is not completely linear. In fact, a 50% increase in NAD(H) pool size compared to untreated controls resulted in a 50 ps decrease in mean NADH lifetime. In comparison, a 50% decrease in NAD(H) pool size resulted in a nearly 150 ps increase in mean NADH lifetime. This is likely because the mean NADH lifetime is limited by the protein bound NADH lifetime of 2500 ps and the free NADH lifetime of 400 ps. Since 70–80% of the NADH is free in the untreated state, further increases in free NADH due to the reduced redox change induced by AA causes the mean lifetime to change less the closer it is to its lower limit of 400 ps. Since NR increases NAD(H) levels, causing the mean NADH lifetime to decrease towards the lower limit, metabolic delta for NR treated cells is consequently lower than that of untreated and FK866 treated cells. These results highlight one limitation of the metabolic delta as an indicator of respiratory differences upon drastic NAD(H) pool size changes.

A further limitation is that the metabolic delta can only eliminate the impact of factors which are stable upon acute AA treatment. Interestingly,

**Fig. 5 | NADH lifetime components allow differentiation between respiration and pool size induced lifetime changes in 143B cells.** τ1 (**a**) and a1/a2 (ratio of free to protein-bound NADH) (**b**) plotted against τmean for mt*Lb*NOX treated cells (*n* = 86) and untreated/antimycin A treated cells (*n* = 20). Correlations and significances were calculated for each group using linear regression. **c** Fold change of FLIM lifetime components (τ1, τ2, a1abs, a2abs, aabs, and a1/a2) normalized to fold change in τmean between 143B cells treated with NR and FK866 (pool size-induced) and between untreated and antimycin A treated cells (respiration-induced) (*n* = 20 for both groups). Each data point is independent. The bars for (**c**) indicate the mean and standard error. Significances were calculated using t tests between selected groups and are indicated as *** for *p* < 0.001.

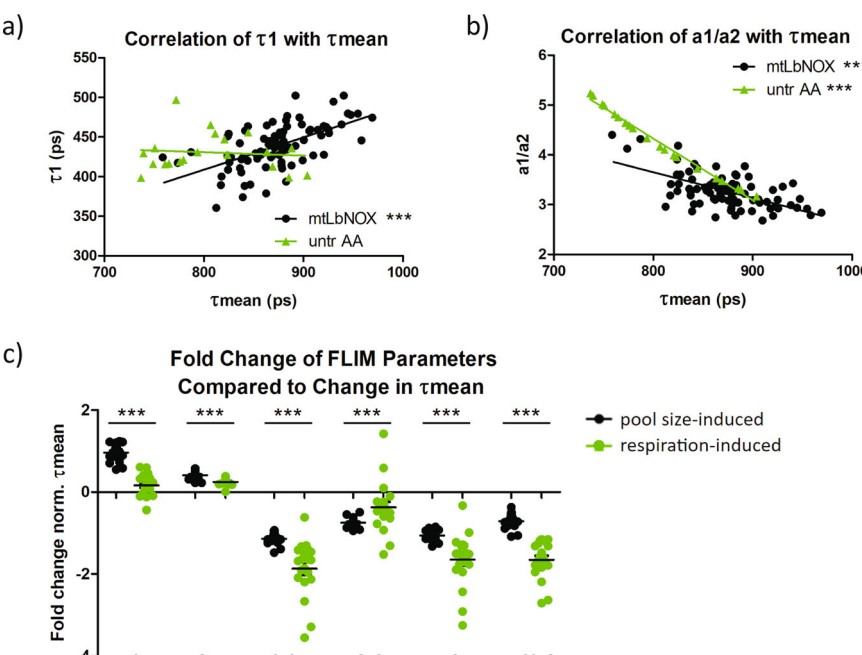

we revealed that the nuclear mean NADH lifetime is significantly increased upon acute AA treatment (Fig. S8b, c), suggesting a more oxidized NAD⁺/NADH ratio. However, this is opposite to what would be expected biologically, since AA should result in an increase in glycolysis and reduces the pyruvate to lactate ratio (Fig. S8d), a known marker for the cytosolic NAD⁺/NADH ratio[24]. A possible explanation is that because antimycin A inhibits respiration and complex I is unable to re-oxidize NADH back to NAD⁺, NAD⁺ is transported into the mitochondria through the NAD⁺ transporter SLC25A51[25]. This would cause a local decrease in NAD(H) pool size in the nucleus, resulting in the longer NADH lifetime upon AA treatment. This emphasizes the critical role of changes in the NAD(H) pool size not only on the cellular level, but also between subcellular compartments. It further demonstrates that subcellular pools of NAD(H) are NOT stable upon acute AA treatment, rendering metabolic delta an imperfect surrogate marker for mitochondrial respiration, although its performance is still superior to using the mean NADH lifetime alone (compare Fig. 4a, c versus Fig. 4b, d).

## Analysis of FLIM components allows differentiation of pool size and respiration induced lifetime changes

To overcome the limitations of the metabolic delta mentioned above, as well as its limited applicability in tissues and in-vivo, we asked if specific changes in individual components of the mean NADH lifetime are associated with pool-size induced or respiration-induced alterations in τmean.

Thus, we re-examined NADH FLIM data from mt*Lb*NOX treated cells and untreated/antimycin A treated cells because the variance in τmean between mt*Lb*NOX treated cells is solely due to differences in the NAD(H) pool size, while the variance in τmean in untreated vs antimycin A treated cells is due to differences in respiration. To assess how the individual lifetime components contribute to changes in τmean, we then plotted them against τmean. We showed that respiration-induced changes in τmean are not associated with changes in the τ1 component (free NADH lifetime) while pool size-induced increases in τmean are associated with a parallel increase in the τ1 component (Fig. 5a). In contrast, respiration-induced changes in τmean are mainly driven by an altered ratio of free (a1) to protein-bound (a2) NADH (a1/a2), while this change in a1/a2 is less prominent for pool size-induced alterations of τmean (Fig. 5b).

To verify these observations upon different treatment conditions, we performed similar analyses in 143B cells between FK866 and NR treatments (pool-size induced) and between untreated and antimycin A treatments

(respiration-induced). We calculated the fold change of each NADH FLIM component normalized to the change in τmean. Similar to our previous results, pool size-induced changes in τmean were associated with a larger fold change in τ1 and τ2 but smaller changes in free (a1abs), protein bound (a2abs) and total (aabs) NADH autofluorescence intensities and the ratio of free to protein bound NADH (a1/a2) compared to respiration-induced changes in τmean (Fig. 5c). This was further validated in HEK293 cells, which also demonstrate that pool-size induced changes in τmean are driven by changes in τ1, while respiration-induced changes in τmean are driven by changes in a1/a2 (Fig. S9). Although the exact mechanism behind these differential changes are not fully elucidated, the consistency of our observations across 2 cell lines and with both chemical (NR/FK866) and enzymatic (mt*Lb*NOX) modifications of NAD(H) pool size, provides strong descriptive evidence for the validity of these observations.

Consequently, individual lifetime components, in particular τ1 and a1/a2, can be utilized to predict whether an observed difference in the mean NADH lifetime between two conditions is due to NAD(H) pool size differences (strong difference in τ1) or due to differences in mitochondrial respiration/redox ratio (strong difference in a1/a2) (Fig. 6). This highlights the potential of NADH FLIM as a tool to evaluate both NAD(H) redox state and pool size changes with subcellular resolution, providing unique insights into redox metabolism in health and disease.

## Discussion
Here, we performed targeted modifications of NAD(H) pool size and redox state and assessed the extent to which fluorescence lifetime imaging microscopy of NADH autofluorescence can detect and distinguish these modifications in cells and tissues. This provided three crucial insights in NADH autofluorescence imaging and its potential in bioenergetic research.

First, we demonstrated that NAD(H) pool size is inversely related to the mean NADH lifetime, so a smaller NAD(H) pool size results in a longer mean NADH lifetime and vice versa. Thus, NADH FLIM can be used to evaluate NAD(H) pool size in cells and tissues when redox state is stable.

While NADH autofluorescence intensity is known to be sensitive to NAD(H) pool size and has previously been used to estimate NAD(H) pool size[16], NADH FLIM relies on the fraction of free to protein-bound NADH and their individual lifetimes τ1 and τ2, which are dependent on the respective microenvironment[9]. Thus, as a ratio-metric readout, one might have hypothesized that NADH FLIM is insensitive to alterations in

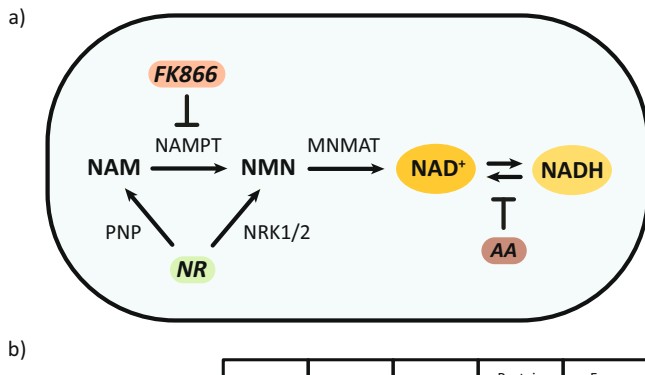

| | | NADH intensity | Mean NADH Lifetime | Free NADH Lifetime | Protein Bound NADH Lifetime | Free: Protein bound NADH |
|---|---|---|---|---|---|---|
| Resp./Redox Changes | Acute Antimycin A | ↑ | ↓ | – | ↓ | ↑ |
| Pool Size Changes | NR | ↗ | ↓ | ↓ | ↘ | ↗ |
| | FK866 | ↘ | ↑ | ↑ | ↗ | ↘ |

**Fig. 6 | Respiration/redox changes and pool size changes differentially impact NADH FLIM parameters. a** NR increases NAD(H) pool size by feeding into the NAD⁺ synthesis pathway, while FK866 decreases NAD(H) pool size by inhibiting the pathway. Antimycin A (AA) alters the NAD⁺/NADH ratio by inhibiting complex III and respiration, preventing the conversion of NADH back to NAD⁺. **b** Summary table outlining the impact of pool size and respiration/redox changes on FLIM parameters. Green up arrows indicate increase while red down arrows indicate decrease. Angled arrows indicate smaller increases or decreases. The indicated changes depict overall tendencies across cell lines and tissues and do not relate to a specific measurement/result.

NAD(H) pool size. However, we demonstrated a causal influence of NAD(H) pool size on the mean NADH lifetime. This can be explained by earlier observations that the protein-bound fraction of NADH is more stable than the free NADH fraction[14]. Thus, an overall decrease in NAD(H) pool size does not cause an equivalent decrease in the free NAD(H) pool compared to the protein-bound NAD(H) pool. Instead, the free NAD(H) pool will show a much larger decrease, thus also affecting the mean NADH lifetime. Consequently, the mean NADH lifetime from NADH FLIM appears to reflect NADH concentration similar to intensity imaging, whether it is affected by changes in redox ratio or NAD(H) pool size.

This raises the question of the additional value of the more technically complex NADH FLIM as opposed to NADH intensity imaging. Most importantly, FLIM imaging is independent of the recorded fluorescence intensity[26], so the results are not affected by differences in absorption or light scattering, which is particularly crucial when imaging deep in tissues or in vivo. Furthermore, by providing the fluorescence intensities of free and protein-bound NADH and their respective lifetimes, FLIM reveals additional information that can be utilized in the biological interpretation of the data, for example for revealing the fraction of NADPH[27] or changes in the proteins which NADH binds to[14].

For evaluation of pool size, compared to the current standard of biochemical assays detecting NAD⁺ and NADH[28], NADH FLIM is non-destructive, and more importantly, allows visualization of pool size differences on a single cell or even subcellular level, as demonstrated by our imaging of mt*Lb*NOX expressing cells (Fig. 2a–c). Similarly, genetically-encoded biosensors, such as SoNar and iNap, also allow subcellular visualization of NAD(P)(H)[29,30]. The advantage of the biosensors is a high specificity for NADH over NADPH (or vice versa) as well as an easier calibration for comparison of absolute redox ratios across cell compartments/cell types. However, NADH FLIM does not require any additional cellular manipulations to express NAD(H) biomarkers, which may be challenging in sensitive or difficult to transfect cell lines. This also means that, unlike genetically encoded biosensors, FLIM can be used directly to quantify NAD(H) pool in tissues, as we have demonstrated in Fig. 3, or even humans[31].

Given the prominent role of alterations in nucleotide levels with aging and disease[32], NADH FLIM can be an invaluable tool in the study of a multitude of disorders. For example, increases in pool size have been associated with colorectal cancer progression[16]. Thus, NADH FLIM can provide a non-invasive method to assess cancer progression[33]. Likewise, NADH FLIM has been used to non-invasively distinguish alterations in cerebral metabolism in live rats[34]. A similar approach can be applied to assess changes in the NAD(H) pool size, for example in neurodegenerative diseases like Alzheimer's disease[35]. This emphasizes the potential of NADH FLIM in estimating alterations in NAD(H) nucleotide levels with subcellular resolution and in vivo.

Second, our finding that NAD(H) pool size impacts the mean NADH lifetime highlights the importance of considering NAD(H) pool size when estimating energy metabolism using NADH FLIM. The most common application for NADH FLIM is assessing energy metabolism and redox state, in particular, the balance of mitochondrial respiration to glycolysis[36]. Since aging and age-associated disorders are associated with metabolic alterations[37], multiple studies have utilized NADH FLIM to assess metabolism upon aging[38,39]. Although mitochondrial function is known to decline with age[40], some studies report a lower free to protein-bound NADH ratio (analogous to a longer τmean) in aged vs. young mice[38]. Traditionally, a longer mean NADH lifetime is interpreted as more oxidative metabolism since it suggests more NAD⁺ relative to NADH[11,41]. However, it is unlikely that aged mice are in fact performing more OXPHOS than young mice. Instead, the longer mean NADH lifetime may be due to a decrease in NAD(H) pool size in tissues of the aged mice, consistent with the reported decline of NAD⁺ biosynthesis with age[42]. This emphasizes that potential changes in the cellular NAD(H) pool size need to be considered to avoid misinterpretations of NADH FLIM results with respect to energy metabolism.

Another scenario when pool size changes might have masked changes in redox state is upon NR treatment. Hu et al. reported that NR causes an increase in cytosolic, but not mitochondrial NAD⁺/NADH ratio[43]. We have observed a significant decrease in NADH lifetime in the mitochondria but not the nucleus/cytosol (Fig. 1a). The more oxidized redox state in the cytosol would increase the NADH lifetime, while the increased NAD(H) pool would decrease the NADH lifetime, resulting in no effect of NR on the cytosolic τmean. In contrast, in the mitochondria the NAD⁺/NADH ratio does not change but the NAD(H) pool is increased, causing a shorter NADH lifetime. This highlights the strong interconnection of both redox state and NAD(H) pool size in altering the NADH fluorescence signal.

Interestingly, the effect of NAD(H) pool size on NADH FLIM does not only come into play in case of systemic alterations in nucleotide levels. Recently, a mammalian mitochondrial NAD⁺ carrier was identified suggesting active redistribution of nucleotides pools between mitochondria and cytosol[25]. Accordingly, it has been shown that the subcellular distribution of NAD(H) changes with cellular differentiation[44] and metabolic state[45]. Thus, the local NAD(H) pool size within cellular compartments can change. Here we have demonstrated how this can result in a misleading interpretation of NADH FLIM. We observed a longer nuclear NADH lifetime following acute antimycin A treatment in both HEK293 and 143B cells (Fig. S8b, c), which normally would be interpreted as a more oxidized nuclear and cytosolic redox state. However, antimycin A blocks respiration thereby increasing glycolysis and shifting the cytosolic NAD(H) redox state towards a reduced state, verified by a reduced pyruvate to lactate ratio. The most likely explanation for a longer nuclear NADH lifetime despite a more reduced redox state is that blockage of the ETC causes nuclear NADH to be shuttled to and trapped in the mitochondria, where it is unable to be re-oxidized. This would cause a local decrease in NAD(H) pool size in the nuclei, accounting for the increased mean NADH lifetime upon AA treatment. Overall, this highlights that the effect of NAD(H) pool size on NADH FLIM always has to be considered when interpreting NADH FLIM data with respect to cellular metabolism or redox state.

Finally, we developed two approaches to distinguish pool size-induced and respiration-induced changes in NADH FLIM. Such separation is important for interpretation of NADH FLIM data because the two scenarios have differing cellular implications. A longer mean lifetime due to changes in respiration suggests increased oxidative phosphorylation and a more oxidized $NAD^+$/NADH ratio. This impacts the balance of all cellular reactions that are dependent on the redox ratio, such as the TCA cycle dehydrogenases. On the other hand, a longer NADH lifetime due to pool size-induced changes suggests an overall decrease in NAD(H) levels, which mostly affects reactions that require $NAD^+$ as a cofactor, such as sirtuins. For example, a longer lifetime due to changes in respiration would indicate more $NAD^+$ relative to NADH and increased (or unaltered) sirtuin activity[46]. On the other hand, a pool size-induced increase in NADH lifetime would suggest decreased $NAD^+$ (and NADH) levels, resulting in lower sirtuin activity. This highlights how a similar effect on the mean NADH lifetime can encode opposing biological underpinnings, emphasizing the need for further interpretation of the mean NADH lifetime.

To isolate respiration-induced changes, we blocked mitochondrial respiration with antimycin A and calculated the difference in NADH lifetime between untreated and respirationally-blocked cells, termed metabolic delta[4]. This allowed correction for all factors which remain stable upon inhibition of respiration. For example, the total cellular NAD(H) pool size does not change with an acute treatment of antimycin A. Thus, a larger cellular NAD(H) pool size will affect the mean NADH lifetimes of untreated and antimycin A -treated cells similarly, rendering the metabolic delta largely insensitive to alterations in the NAD(H) pool size.

However, this approach has two limitations. First, we demonstrated that τmean is not linearly correlated with NAD(H) pool size when τmean approached the free and protein bound lifetime limits of 400 ps and 2500 ps. Since the fractions are usually shifted towards free NADH[41], this results in slightly lower metabolic deltas when the overall NADH lifetimes are shorter. Accordingly, we observed a slightly lower metabolic delta in NR treated cells compared to FK866-treated cells, although mitochondrial respiration was largely comparable.

The same applies for the correlation of τmean with respiration. However, metabolic changes only induce 100-200 ps differences in τmean when factors such as pool size and pH are controlled for. Thus, the relationship between τmean and respiration is more likely to be monotonic in the relevant ranges[10].

Second, we demonstrated that, while acute antimycin A treatment does not alter total cellular NAD(H) pools, it shifts the intracellular NAD(H) localization away from the nucleus and towards the mitochondria. Thus, when performing subcellular analysis, intracellular NAD(H) pool size may not be stable and can impact FLIM results. Therefore, the metabolic delta provides a decent but still imperfect estimate of the contribution of respiration-induced NADH lifetime changes.

Our second approach was based on the hypothesis that individual parameters of NADH FLIM are differentially sensitive to respiration-induced versus pool size-induced alterations in the mean NADH lifetime. Indeed, we could demonstrate that respiration-induced lifetime alterations are mostly due to changes in the ratio of free to protein-bound NADH (a1/a2), whereas pool-sized induced alterations show significantly higher changes in τ1 and partially in τ2. One underlying reason for these different characteristics could be that changes in the NAD(H) pool size affect NADP(H) pool size to a lesser extent, altering the ratio of NADPH to NADH. It has been reported that protein-bound NADPH has a longer lifetime compared to NADH[15]. Thus, the change in the NADPH to NADH ratio could explain the influence of τ2 on pool-size induced lifetime alterations. However, the free NADPH lifetime is identical to free NADH lifetime[15], rendering it unlikely that alterations in NADPH/NADH are the main contributing factor for the overproportional changes in τ1 with alterations in NAD(H) pool size. Since the lifetime components τ1 and τ2 are sensitive to the microenvironment, such as viscosity and pH[14,47], we hypothesize that changes in pool size result in a different subcellular distribution of NAD(H) compared to changes in respiration, causing differential changes in the cellular microenvironment. This results in specific changes in FLIM parameters, allowing differentiation of respiration-induced NADH lifetime alterations from pool size-induced alterations. Although the exact underlying molecular mechanisms behind these observations are not fully elucidated, they are consistent across 143B cells and HEK293 cells, as well as with 2 different methods of altering NAD(H) pool (chemically via NR/FK866 and enzymatically via mt$Lb$NOX), providing strong descriptive evidence that the differential changes are genuine. However, it is possible that our findings may not be corroborated in different cell types or in tissues, thus, more work in the future is needed to determine whether FLIM parameters behave similarly across cell types and tissues upon targeted modifications to respiration or NAD(H) pool size. Despite this, our work depicts an important stepping stone in first identifying the NAD(H) pool size as a factor that should be considered in NADH FLIM interpretations and providing a working hypothesis to differentiate the influencing factors.

The differential changes in FLIM parameters with pool size vs respiration changes also inform additional methods for analyzing NADH FLIM data depending on the selected application. For example, when detection of respiration changes is desired, metabolic delta can be calculated using the ratio of free to protein-bound NADH (a1/a2) rather than τmean, as we show that a1/a2 is less sensitive to pool size alterations than τmean[48]. Thus, the resulting values are less likely to be confounded by any pool size changes.

Overall, given that NADH autofluorescence is only a surrogate marker, additional measures may be needed to ensure its accurate interpretations with respect to energy metabolism, redox state and NADH pool size. Firstly, NADH FLIM of antimycin A treated samples enables correction for a range of different factors, including NAD(H) pool size to some extent, and can isolate changes in respiration between samples. Next, when using FLIM as a direct measure of mitochondrial respiration, a matrix pH sensor may need to be simultaneously imaged to correct for pH mediated redox changes, as matrix pH tends to fluctuate greatly[10]. Finally, to ensure that NAD(H) pool size is not a confounding factor, separate samples can be saved for biochemical quantification of total NAD(H). However, because this does not address cases of re-distribution of NAD(H) within the cell, fold change analysis of the FLIM parameters can optionally be performed to further distinguish between respiration-induced and NAD(H) pool size-induced lifetime changes.

In summary, we reveal that NADH FLIM is sensitive to changes in the levels of $NAD^+$ and NADH, promoting NADH FLIM as a valuable tool to estimate NAD(H) pool size optically. At the same time, our findings reveal NAD(H) pool size as a crucial parameter which needs to be considered when performing NADH FLIM to evaluate cellular energy metabolism and mitochondrial function. Despite this, we are able to use additional analytical approaches to estimate mitochondrial respiration, redox state, and NAD(H) pool size with subcellular resolution, highlighting the potential of NADH FLIM as a crucial tool in aging and bioenergetics research.

## Methods
### Mouse strains and NR treatment
All mice used in this study are male on the C57BL/6[Eij] (Nnt + /+) background ("B6" controls). In addition, mice deficient for the adenine nucleotide translocator 1 (ANT1, Slc25a4-/-)[49] were used that further harbor the complex I variant ND5 m.12352 C > T (ND5[S204F]). All mice were housed in a barrier facility, fed a low-fat 5L0D diet and kept on a 12 h light/dark cycle. Only male mice between 6 and 24 months of age were used in this study. Since the goal of the study is the methodological establishing of NADH FLIM in liver tissue and not the underlying biology, we did not assess both genders.

For NR treatment, NR (Nicotinamide Riboside chloride) powder (Chromadex) was dissolved in in water (15 mg/ml) and sterile filtered on the day of the experiment. The NR solution or 0.9% saline solution (sham) were IP-injected into the mice (100 μl/30 g of mouse weight) to achieve a final NR concentration of 500 mg/kg body weight. Animals were sacrificed by

cervical dislocation 3 h post injection, the right liver lobe was dissected, parts of it flash frozen and stored at –80 °C for biochemical analysis and one part imaged using NADH FLIM.

The Institutional Animal Care and Use Committee from the Children's Hospital of Philadelphia approved all animal protocols of this research. We have complied with all relevant ethical regulations for animal use.

## Cell culture

143B(TK-) osteosarcoma cells and HEK293 cells were maintained in T75 cell culture flasks in 10 ml of Dulbecco's modified medium (DMEM) from Gibco (10569-010) supplemented with 10% fetal bovine serum and 2 mM uridine at standard cell culture conditions (37 °C, 5% $CO_2$). 143B(TK-) cells were obtained from the Wallace laboratory collection of cell lines, and HEK293 cells obtained from the American Type Culture Collection. Cell identities were confirmed by karyotype analysis and cell lines routinely screened for mycoplasma using commercial PCR kits. Cells were split every 2-3 days. For the experiments, cells were seeded 48 h prior and the medium was changed 24 h prior to the experiment. NR and FK866 treatments were administered with the fresh media. NR-containing media was prepared fresh every week to exclude a change in concentration due to possible degradation.

## Oroboros

High-resolution respirometry was performed using the Oroboros Oxygraph-2k in intact cells to mirror the energy metabolic state (endogenous substrates) during NADH FLIM[10]. Oroboros chambers were first calibrated to room air using 2 mL of cell culture media (as described above). Conditioned media from the cells was collected and spun to remove cells and debris (1000 g, 3 min). Cells were washed with PBS, trypsinized and resuspended in their conditioned media at a concentration of 1 million cells per mL. 2 million cells were loaded into each chamber, and the oxygraph signal was allowed to plateau. This plateau represented routine respiration. Subsequently, 0.5 μL of 10 μM oligomycin was added to block Complex 5 activity and determine leak respiration. Next, 1 μL of 1 mM FCCP was added to uncouple the inner mitochondrial membrane and simulate maximal ETC activity. After the signal plateaued, additional FCCP was added in 0.5 μL increments until no further increase in respiration was observed. This final plateau corresponded to ETS capacity. Finally, 2 μL of 5 mM antimycin A was added to inhibit complex 3 and measure background or non-mitochondrial oxygen consumption.

## Glycolysis and doubling time

Conditioned media was collected in the context of the Oroboros measurements and stored at –80 °C until lactate levels were measured using the Glycolyis Cell-Based Assay Kit (Cayman Chemicals). Lactate levels were normalized to the collection time and average cell number during the collection time to reveal the lactate secretion rate[50].

Doubling time was calculated by counting cells prior to plating and again when harvesting the cells for the Oroboros experiments. The formula *Doubling Time (hrs)* = [*Culture Time (hrs)* / [$\log_2$ (*Cell Number* $_{end}$ / *Cell number* $_{start}$)]] was used to quantify doubling time.

## Seahorse metabolic flux assay

Cells were seeded in Seahorse XF96 Cell Culture Microplates at 36000 cells/well for HEK293 cells and 17000 cells/well for 143B cells 24 h prior to the experiment to achieve around 70–80% confluence on the day of the experiment, and treatments (NR, FK866) were administered simultaneously in 100 μL of the culture media (described above). 4 wells were left unseeded in each experiment to serve as background measurements. One hour prior to performing the experiment, cell culture growth media was replaced by unbuffered Seahorse XF DMEM media (pH 7.4) supplemented with 2 mM glutamine, 1 mM sodium pyruvate, and 5 mM glucose and cell plates subsequently incubated in a low $CO_2$ incubator for around 45 min to equilibrate pH. To evaluate mitochondrial oxygen consumption rates (OCR) and extracellular acidification rates (ECAR), the XF Mito Stress Test

was performed according to the manufacturer's instructions on the Seahorse XF96 Analyzer. Basal respiration was obtained by measuring at 3 consecutive time points prior to any pharmacological perturbation. OCR and ECAR was averaged across all three time points to give a single measure for each well.

## mt*Lb*NOX experiments

Plasmids were designed and ordered from ATUM. mt*Lb*NOX[23] and a red fluorescence reporter (CayenneRFP) were designed to be under the control of a CMV promoter with a T2A site to allow them to be transcribed as one transcript and later separated during translation. Inclusion of a neomycin resistance gene enabled selection of transfected cells.

143B osteosarcoma cells were plated and transfected at 30% confluence using TransIT-X2 after 24 h. Successfully transfected cells were selected for by addition of 400 μg/ml G418 for 2 weeks and continuous culture under 200 μg/ml G418. Clonal selection was not performed to generate a cell line with heterogeneous mt*Lb*NOX expression.

CayenneRFP fluorescence was measured at the same time as NADH FLIM, allowing correlations between mean NADH lifetime and CayenneRFP fluorescence (marker for mt*Lb*NOX expression) in each cell. CayenneRFP fluoresence in individual cells was quantified with Fiji (ImageJ) by first marking a region of interest around each cell with the "Freehand Selection" tool and then measuring average pixel intensity. Corresponding cells were analyzed in NADH FLIM images for mean lifetime as described in subsequent sections. The mean lifetime of each cell was normalized to the average τmean of all untreated or antimycin A treated cells, respectively.

## NADH FLIM of cells

Cells were seeded in 35 mm dishes with glass bottom (Greiner, 627870) 48 h prior to the experiment in standard cell culture medium (as above) supplemented with 25 mM HEPES. NR/FK866 treatments NR/FK866 treatments of varying concentrations (300 μM and 1 mM NR; 1 nM, 1.5 nM, 2 nM, and 5 nM FK866) were administered in the same medium 24 h prior to the experiments. Directly before the experiment, media was removed and cells were imaged in Tyrodes buffer (135 mM NaCl, 5 mM KCl, 1.8 mM $CaCl_2$, 20 mM Hepes, 5 mM glucose, 1 mM $MgCl_2$, pH 7.4). Antimycin A treatments (5 μM), used to block respiration, were administered directly prior to measurements.

Fluorescence lifetime imaging of NADH was performed on a laser scanning microscope (Zeiss LSM 710) as described previously[51]. Briefly, NADH was excited using two-photon excitation with a pulsed (80 MHz, 100 fs pulse width) titanium-sapphire laser at a power of <1.5 mW on the sample. Time-correlated single photon counting (TCSPC) with the hybrid detector HPM-100-40 was performed. The detector was coupled to the NDD port of the LSM 710. FLIM images (512 × 512 pixel) were taken at a temporal resolution of 256 time channels. Final settings: 60 sec collection time; ≈ 15 μsec pixel dwell time; 135 × 135 μm$^2$ scanning area, Plan-Apochromat 63x/1.40 Oil DIC M27 lense, 730 nm excitation wavelength and 460/50 nm bandpass emission filter. Data were recorded using SPCM 9.8 and subsequently analyzed using SPCImage 8.8 for all cytoplasmic data and SPCImage 8.0 for all other data assuming a biexponential decay.

Standard analysis settings if not indicated otherwise (Figs. 1, 2, 3 and Fig. S1, S3): WLS fit method, biexponential decay with unfixed lifetime components (τ1 and τ2), square binning of 2, peak threshold adapted to background, shift fixed at a pixel of clear NADH signal. The mean lifetime (τmean) was calculated and fitting of the calculated lifetime curve was confirmed by checking the mean χ2, which should be below 1.2. For subcellular analysis, nuclei were selected by drawing regions of interest (ROIs) using the "Define mask" tool in areas clearly distinguishable as nuclei by eye. At least five nuclei were masked for analysis per image. The mitochondria-rich regions were isolated from the nucleus and cytoplasm using the intensity threshold function because mitochondria tend to have higher fluorescence than the surrounding cytoplasm and the nucleus. The cytoplasmic regions were selected for quantification by marking a region of interest using the "Define Mask" tool in areas clearly distinguishable as

lacking mitochondria by eye. Cytoplasmic regions from at least five cells were selected per image. 2D correlations in SPCImage 8.8 were used to isolate cytoplasmic regions for visualization in representative images by limiting τmean to exclude the shorter nuclear lifetimes on one axis and limiting a1abs to exclude high intensity mitochondrial regions on the second axis. For single cell analysis of nuclear, mitochondrial, and cytoplasmic τmean, the "Define Mask" tool was used to mark a region of interest around the selected cell or nucleus.

Metabolic Delta: Individual data points for metabolic delta were obtained by subtracting the τmean from each individual image for each treatment group (untr, NR, FK866) after antimycin A treatment from the average τmean of each treatment group before antimycin A treatment.

FLIM parameter analysis: Since mt*Lb*NOX and antimycin A treatments produced a continuous range of alterations in τmean, unfixed lifetime components (τ1 and τ2) allowed direct correlation of the lifetime parameters, displayed for τ1 and a1/a2, with τmean.

Pool-size induced lifetime alterations were calculated as lifetime component $x_{FK866}/x_{NR}-1$ and subsequently normalized to respective alteration in τmean between the same images calculated as $\tau mean_{FK866}/\tau mean_{NR}-1$. Accordingly, the respiration-induced lifetime alteration was calculated as $(x_{untr}-x_{AA}-1)/(\tau mean_{untr}/\tau mean_{AA}-1)$ with x being any of the aforementioned individual lifetime components.

## NADH FLIM of liver

Liver tissue was dissected from mice sacrificed by cervical dislocation and immediately placed in Tyrodes buffer. Tissue was then mounted on an imaging chamber flooded with Tyrodes buffer and NADH FLIM was performed as described above but with a lower laser power <1.5 mW on the sample and using a 20x air lens (NA 0.8) with a 425.1 × 425.1 μm² scanning area. Due to high amounts of non-NADH autofluorescence in liver NADH FLIM images, the phasor plot was calculated in SPCImage. The phasor plot does not require an assumption of the number of lifetime components or a fitting procedure but instead displays the lifetime of each pixel as a phasor within a half-circle[52]. Pixels directly on the half-circle are monoexponential, with pixels to the right representing a shorter lifetime. Pixels within the half-circle are multiexponential. Since the phasor plot does not require a fitting procedure, it is ideal to distinguish a mix of autofluorophores according to their lifetime characteristics. The region with NADH signal was identified by common NADH autofluorescence τ1 and τ2 values (the non-NADH autofluorescence showed a very long τ1 ~ 1000 ps) and validated by comparing phasor plots of control tissue with phasor plots of antimycin A treated tissue. Antimycin A only impacts NADH fluorescence, causing a shift in the location of the NADH signal region in the phasor plot.

The pixels with an NADH autofluorescence signal profile were selected using a region of interest (ROI) by checking "Select Cluster". The photons of all pixels within the ROI were summed up and the biexponential decay was calculated with non-fixed lifetime components and non-fixed shift.

## NAD⁺/NADH and NADP⁺/NADPH quantification

Quantification in cells: For biochemical quantification of NAD⁺, NADH and NADP⁺, NADPH using the NAD/NADH-Glo™ Assay and the NADP/NADPH-Glo™ Assay (Promega), the provided protocol was followed. Briefly, cells were seeded 48 h prior and respective treatments were performed 24 h (NR, FK866) or 5 min (antimycin A) prior to the experiment. The media was removed, cells were washed in ice-cold PBS and lysed in 100 μl base solution (0.2 N NaOH + 1% DTAB). Samples were split in half for measuring reduced or oxidized nucleotides respectively. For the detection of NADH, samples were heated to 60 °C for 15 min and for the detection of NAD⁺ samples were first acidified by adding 25 μl of 0.4 N HCl and subsequently heated to 60 °C for 15 min. Following neutralization using HCl-Trizma or Trizma base respectively, the assay kit was performed and luminescence recorded.

For quantification in subcellular fractions, mitochondrial isolation was performed by first washing cells in ice cold PBS and collecting using a cell scraper. Cell samples were then resuspended in 1 mL of ice cold mitochondrial isolation buffer (75 mM sucrose, 225 mM mannitol, 10 mM MOPS, pH 7.4)[53]. EGTA was not included in the buffer to prevent interactions with the NAD or NADP cycling enzyme in subsequent steps. All subsequent steps were performed on ice. Cells were lysed with 20 strokes of a pre-chilled glass Dounce homogenizer, lysate centrifuged at 1000 g, and resulting supernatant centrifuged again at 1000 g to remove unbroken cells and nuclei. The mitochondrial fraction was obtained by centrifuging at high speed (10000 g). The supernatant after high-speed centrifugation was collected as the cytosolic fraction, and the remaining mitochondrial pellet was washed in 1 mL mitochondrial isolation buffer and collected by spinning once more at 10000 g. Mitochondrial pellets were resuspended in 110 μl mitochondrial isolation buffer. Protein was quantified in mitochondrial and cytosolic fractions using the Pierce™ BCA protein assay (Thermo Scientific) according to manufacturer's protocol for normalization. 50 μl of each cytosolic and mitochondrial sample were used to quantify NAD⁺, NADH and NADP⁺, NADPH using the NAD/NADH-Glo™ Assay and the NADP/NADPH-Glo™ Assay (Promega) as described above, using 50 μl of base solution for lysis instead.

For biochemical quantification of NAD⁺ and NADH in control and mt*Lb*NOX cells using the NAD/NADH Quantification Kit (Sigma-Aldrich), the provided protocol was used. Cells were seeded 24 h prior to the experiment. On the day of the experiment, media was removed and cells were then washed in PBS, trypsinized, and counted for 20,000 cells per sample. Each sample was lysed in 400 μl of provided extraction buffer. Half of the cell lysate was separated and heated to 60 °C for 30 min to degrade NAD⁺. NAD(H) and NADH were then detected colorimetrically according the manufacturer's protocol.

Quantification in tissue: For biochemical quantification of NAD⁺, NADH and NADP⁺, NADPH in liver the NAD/NADH Quantification Kit (Sigma-Aldrich) was used and an NAD(H) cycling assay was performed. Upon cervical dislocation, pieces of the right liver lobe were dissected and flash frozen. The liver pieces were powdered on dry ice and the tissue powder was portioned into tubes and weighted.

For the quantification kit, liver powder was dissolved in the provided extraction buffer (200 μl/10 mg of tissue powder) and incubated on ice for 15 min. Afterwards the lysate was spun and the supernatant split into to two tubes. One was used for NAD(P) + NAD(P)H detection and one was heated to 60 °C for 30 min to detect NADH only. Subsequently the NAD(P)(H) detection kits were performed following the manufacturers protocol.

For the cycling assay, two separate tubes were used. For NADH the powdered tissue was lysed in ice-cold 0.25 M KOH in 50% ethanol while for NAD⁺ it was lysed in ice-cold 0.6 M Perchloric Acid. Samples were spun down for 15 min at 20,000 g at 4 °C, the supernatant transferred to new tubes and diluted 1:40 (1:25 for NADH) in ice cold Na-Phosphate buffer (pH 8.0). Diluted extracts or standards were added to a cycling mixture (2% ethanol, 100 μg/mL alcohol dehydrogenase, 10 μg/mL diaphorase, 20 μM resazurin, 10 μM flavin mononucleotide, 10 mM nicotinamide, 0.1% BSA in 100 mM phosphate buffer, pH 8.0) and fluorescence was quantified at 590 nm (ex:530 nm) in a 96 well plate.

## Targeted LC/MS metabolomics

Approximately 5 mg aliquots of dry liver powder were homogenized in 500 μl of 80% ice cold methanol in the Precellys homogenizer at 4 °C. Then, 100 μl of homogenates were extracted with 400 μl of ice-cold methanol, vortexed, centrifuged at $18,000 \times g$, and 400 μl aliquots of supernatants were dried under nitrogen at 45 °C in a 96-well plate. An Agilent PEEK poroshell HIILIC-z column was used to separate nucleotides with gradient elution. Quantitation of metabolites in each assay module was achieved using multiple reaction monitoring of calibration solutions and study samples on an Agilent 1290 Infinity UHPLC/6495 triple quadrupole mass spectrometer. Raw data were processed using Mass Hunter quantitative analysis software (Agilent). Calibration curves (R2 = 0.99 or greater) were either fitted with a linear or a quadratic curve with a $1/X$ or $1/X^2$ weighting.

## Statistics and reproducibility

Each point in graphs represents an independent data point with the line indicating the mean and the error bars indicating the SEM. Data were tested for Gaussian distributions using the D'Agostino and Pearson omnibus normality test. If all data passed the normality test ($p > 0.05$), an unpaired two tailed t-test (two groups) or a One-Way ANOVA with Bonferroni correction for multiple testing was performed. If any data did not pass the normality test, a Mann-Whitney test (two groups) or a Kruskal-Wallis test with Dunns' correction for multiple testing was performed.

To elucidate the effect of NR in-vivo across mouse strains and age, 6- and 18-month-old mice of B6 and ANT1 mice were used. To reveal the effect of NR, each independent data point was normalized by dividing it by the mean value for the respective saline-injected control group.

A correlation between mean NADH lifetime and NAD⁺/NADH ratio was generated by correlating the averaged value for mean lifetime of each treatment group (300 μM NR, 1 mM NR, untr, 1 nM FK855, 1.5 nM FK866, 2 nM FK866, and 5 nM FK866) with the averaged value for biochemically measured NAD⁺/NADH ratio. The correlation between mean NADH lifetime and NAD(H) levels was similarly calculated.

Significances between treatment groups (*), between antimycin A treated and untreated groups (#), or between mitochondrial and cytosolic fractions (#) are displayed as n.s. $P > 0.05$, $*/\# \ p < 0.05$, $**/\#\# \ p < 0.01$, $***/\#\#\# \ p < 0.001$.

To ensure reproducibility, NADH FLIM imaging of untreated (HEK293, 143B, and mtLbNOX) and corresponding treated (NR, FK866, AA) cells was repeated in five independent experiments on separate days, and five images taken from different areas of each treatment well during each experiment. Quantification of NAD(P)(H) was performed using several methods, including by two types of commercially available biochemical kits and by cycling assay, to ensure data reproducibility.

Sample sizes for all experiments were determined based on similar experiments in the field. Sample size was maximized where possible and in accordance with experimental bandwidth. Technical replicates were averaged and biological replicates were treated independently. A minimum of three biological replicates were used for all quantifications, and a minimum of 3 independent experiments were performed for all respirometry data.

## Data availability

The authors declare that all data supporting the findings of this study are included in the paper, its Supplementary Information, and its Supplementary Data.

## Code availability

The authors declare that no new code was created or used to analyze the data.

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

## Acknowledgements

We would like to thank Christopher Petucci and the Metabolomics Core in the Cardiovascular Institute at the University of Pennsylvania for performing and analyzing the global and targeted metabolomics experiments. The authors would like to express their gratitude to Niagen® for providing the NR. Lastly, the authors would like to thank Piotr Kopinski for his assistance in creating the mt*Lb*NOX plasmids. This work was supported by the German Research Foundation (SCHA 2182/1-1) and the Foerderer Award 2020 awarded to PM Schaefer, the National Institutes of Health grants NS021328, MH108592, OD010944 and U.S. Department of Defense grants W81XWH-16-1-0401 and W81XWH-21-1-0128 (PR202887.e002) awarded to DC Wallace.

## Author contributions

P.M.S. conceptualized the project and designed research; A.S., N.Z, C.P. and P.M.S. performed the experiments; A.S., D.C.H. and P.M.S. analyzed data; A.S., J.A.B., D.C.W., and P.M.S. interpreted the data, A.S., D.C.W., and P.M.S. wrote the paper. All authors read, edited and approved the manuscript.

## Competing interests

The authors declare the following competing interests: J.A.B. is consultant to Pfizer and Cytokinetics, an inventor on a patent for using NAD+ precursors in liver injury and has received research funding and materials from Elysium Health and Metro International Biotech, both of which have an interest in NAD+ precursors. D.C.W. is part of the scientific advisory boards for Pano Therapeutics and Medical Excellence Capital. All other authors declare no competing interests.
