## [Peer review file · Communications Biology]

Reviewers' comments:

Reviewer #1 (Remarks to the Author):

Song et al. evaluated the compartmental NADH FLIM in cells and liver and visualized NAD(H) pool size differences on a single cell level. They also built up a method that could separate respiration-induced or non-respiration-induced NADH changes, but there were two limitations for metabolic delta. At last they employed mtLbNOX or chemicals to analyze whether NADH FLIM could be a tool to distinguish NAD(H) redox state and pool size changes at subcellular resolution. Although they also used some another enzyme method to support their results, there are still some concerns.

- (1) The author will also use Seahorse to measure energy metabolism to support their data.
- (2) Since NADPH autofluorescence cannot be distinguished from NADH fluorescence spectrally, the authors will employ genetically encoded indicators to demonstrate the results, such as Frex, iNap or Appollo-NADP+ (PMID: 21982715, 28581494, 26878383).
- (3) In Mootha's paper, they did not show LbNOX change NAD(H) pool size (PMID: 27124460). But in this manuscript, the author showed mtLbNOX may change the NAD(H) pool size. LbNOX is from H₂O-forming NADH oxidases, the author should explain how does it change NAD(H) pool size.
- (4) In this study, the authors found that NR increased NAD⁺ and NADH levels, but the NAD⁺/NADH ratio may not be changed. But in Wang's paper, they found NAD⁺ precursor NR or NMN increased cytosolic NAD⁺/NADH ratio, not mitochondrial NAD⁺/NADH ratio (PMID: 34901920). So the author should describe it precisely, NR did not change mitochondrial or nucleus NAD⁺/NADH ratio, and talked about these results in the discussion.
- (5) The authors did not describe details very clearly. For example, what is the concentration of NR or FK866 for NADH FLIM in cell? The glycolysis contributes NADH in cytosol. Why did the author focus on mitochondria and nucleus, not cytosol? How did the author use NADH FLIM to measure subcellular NADH? What are the targeted modifications?

Minor

- (1) It is NAD⁺ not NAD⁺. It is mtLbNOX, not mtLBNOX.

Reviewer #2 (Remarks to the Author):

This work demonstrated the distinctive impacts of changes in NAD(H) redox state and NAD(H) pool size on FLIM parameters of NADH. The authors showed that both energy metabolism perturbations and changes in NAD(H) pool size impact the mean fluorescence lifetime of NADH, but they have different impacts on the short and long fluorescence lifetime and the free-to-bound NADH ratio, enabling the distinctive characterization of changes in energy metabolism and nucleotide pool size from non-invasive FLIM measurement of NADH. This work also cautions about the interpretation of NADH FLIM measurements given these two distinctive contributions. This is one of a rare study that systematically explores distinct contributions to FLIM of NADH, especially the effect of NAD(H) pool size that is mostly unaccounted for in the interpretation of FLIM measurements. Hence, I think this is an important contribution to the metabolic imaging field. To help clarify and improve the interpretation of the experiment, I have the following comments:

Major comments:

1. The authors segmented the NADH compartments into mitochondria and nucleus, but what about cytoplasm where glycolysis happens?
2. mtLbNOX is used to perturb NAD(H) pool size and is shown to leave NADH redox state unchanged. This is a bit counter intuitive to me because mtLbNOX is an NADH oxidase, and one would expect the exact opposite. Is there an argument for using mtLbNOX to perturb the NAD(H) pool size?
3. NADH can also bind to mtLbNOX, which can potentially change its fluorescence lifetime. How to make sure the observed change of NADH fluorescence lifetime is not due to this potential confounding factor?
4. The metabolic delta is a very interesting parameter that can potentially isolate the change of respiration from the change of NAD(H) pool size, but is mean fluorescence lifetime the best choice to compute this quantity? What about the free-to-bound NADH ratio, which is more sensitive to respiration perturbation and is potentially free of the artifacts suffered from the mean lifetime as explained in the manuscript?
5. The mechanism how respiration perturbation and NAD(H) pool size perturbation impacts FLIM parameters distinctively remains elusive to me. How does NAD(H) pool size change short NADH lifetime more than respiration, given both perturbations could change pH or other microenvironment factors?

Minor comments:

6. How to test that respiration perturbation impacts redistribution of NAD(H) across subcellular compartments?
7. In Figure 2a), AA does not seem to change the mean lifetime, is this as expected?
8. In the introduction, it is claimed that protein-bound NADH remains relatively stable, but Figure 5 and S6 seem to show significant change in this quantity?
9. Following question 8, is the correlation between mean NADH lifetime and respiration activity always monotonic (after excluding the change of NAD(H) pool size)?

Reviewer #1 (Remarks to the Author):

(1) The author will also use Seahorse to measure energy metabolism to support their data.

As suggested by the reviewer, we performed Seahorse assays on 143B and HEK293 cells treated with the highest concentrations of NR (1mM) and FK866 (5nM). Whereas the oxygen consumption rate (OCR) was not affected in 143B cells, the 5nM FK866 treatment significantly lowered respiration in HEK293 cells. This effect was only observed at the highest FK866 concentration, and is not totally surprising, since at 5nM FK866, NAD(H) levels are almost undetectable in cells (Fig.1c). Analysis of OCR/ECAR also revealed no differences upon either treatment in 143B cells, however, OCR/ECAR was significantly lower at many concentrations of FK866 in HEK293 cells. Thus, FK866 slightly shifts energy metabolism in HEK293 cells towards glycolysis. However, these changes cannot explain the NADH lifetime changes that we observe with FK866 because a smaller OCR or a smaller OCR/ECAR ratio should cause a more reduced redox ratio, which would shorten the NADH lifetime. Instead, we observed significantly increased lifetime upon FK866 treatments, recapitulating our hypothesis that changes in NADH fluorescence lifetime upon FK866 treatment are due to changes in NAD(H) pool size rather than changes in cellular metabolism.

We have added the following figures and corresponding discussion to our manuscript as **Figure S3** and as follows (**page 4**, line 1):

“Finally, to additionally confirm neither NR nor FK866 impact metabolism, we performed Seahorse assays on 143B and HEK293 cells treated with 1mM NR and 5nM FK866. Oxygen consumption rate (OCR) was not effected in 143B cells (Fig.S3a), but 5nM FK866 treatment significantly lowered OCR in HEK293 cells (Fig.S3b,c). This was only observed at the highest FK866 concentration, and is not totally surprising given that that 5nM FK866, NAD(H) levels are nearly undetectable (Fig.1c). The OCR/ECAR ratio was not impacted by either treatment in 143B cells (Fig.S3d), but was decreased in HEK293 cells treated with FK866 (Fig.S3e,f). Thus, FK866 treatment shifts metabolism in HEK293 cells towards glycolysis. However, this cannot explain the lifetime changes we observe, because a decreased OCR/ECAR ratio suggests a more reduced redox ratio, which would shorten the NADH lifetime. Instead, we observe significantly increased lifetime upon FK866 treatments, recapitulating that the altered lifetimes upon NR and FK866 treatment are not caused by differences in cellular energy metabolism.”

Figure S3. Seahorse assays show that lifetime changes upon NR and FK866 treatment are not due to changes in metabolism.

a/b) Baseline OCR as measured by Seahorse assay in 143B cells (a) and HEK293 cells (b) treated with NR and FK866. **c)** Baseline OCR as measured by Seahorse assay in HEK293 cells with FK866 titration. **d/e)** Baseline OCR/ECAR as measured by Seahorse assay in 143B cells (d) and HEK293 cells (e) treated with NR and FK866. **f)** Baseline OCR/ECAR as measured by Seahorse assay in HEK293 cells with FK866 titration. $n = 3$ independent experiments for all Seahorse assays. The bars indicate the mean and standard error. Significances were calculated using ANOVA and Dunn's post hoc test between selected groups and are indicated as n.s. for not significant, * for $p < 0.05$, ** for $p < 0.01$, and *** for $p < 0.001$.

(2) Since NADPH autofluorescence cannot be distinguished from NADH fluorescence spectrally, the authors will employ genetically encoded indicators to demonstrate the results, such as Frex, iNap or Appollo-NADP⁺ (PMID: 21982715, 28581494, 26878383).

We highly appreciate the suggestion to measure NADH and NADPH separately using genetically encoded biosensors. We agree with the reviewer that NADPH could potentially confound our NAD(P)H autofluorescence measurements. To ensure that NADPH did not bias our results, we have quantified NADH and NADPH via three independent methods (NAD(P)/NAD(P)H Kit and NAD(P)(H) cycling assay) in two different laboratories. This demonstrated no differential effects of our treatments on NADPH versus NADH except for after FK866 treatment (Fig.S4c), in which the effect on NADH is much stronger than on NADPH. Thus, our results demonstrate that on the whole-cell level, NADPH is not a major confounding

factor in our experiments. As detailed in our discussion, it is well possible that upon a decreasing NADH pool size, the increased proportion of NADPH autofluorescence is responsible for the longer lifetime components, in particular τ_2 . To further elucidate possible differential effects of NADPH versus NADH on a subcellular level, we have added the analysis developed by Blacker et al., 2014. This demonstrates that our quantification of the NADPH/NADH ratio using biochemical assays is consistent with their method to determine the NADPH/NADH ratio using fluorescence lifetimes, reiterating that NADPH is likely not a major confounding factor.

We have added the following figures as **Figure S4d-g** and modified our manuscript as follows (**page 4**, line 18) to include the analysis developed by Blacker et al.:

“To verify our biochemical quantifications, we also performed analysis developed by Blacker et al. to determine the NADPH/NADH ratio in treated and untreated cells using lifetime components from FLIM [15] (*Fig.S4d,e*). This analysis showed close correlations between the biochemically and optically quantified NADPH/NADH ratio in both mitochondria and the nucleus, suggesting no significant differential effects on the subcellular level (*Fig.S4f,g*).”

d/e) NADPH/NADH ratio in the nucleus (d) and the mitochondria (e) quantified using FLIM parameters by Blacker et al.'s methods [15]. **f/g)** Correlation between biochemically quantified NADPH/NADH and NADPH/NADH as quantified using Blacker et al.'s method [15] in the nucleus (f) and the mitochondria (g). Correlations were calculated using linear regression.

(3) In Mootha's paper, they did not show LbNOX change NAD(H) pool size (PMID: 27124460). But in this manuscript, the author showed mtLbNOX may change the NAD(H) pool size. LbNOX is from H₂O-forming NADH oxidases, the author should explain how does it change NAD(H) pool size.

Thank you very much for this comment. We must admit that we were similarly surprised about this result, as we initially intended to employ mtLbNOX to alter the NAD⁺/NADH ratio, not the pool size. However, as indicated above, multiple, independent methods to quantify NAD⁺/NADH verified that in these cell types, mostly the NAD(H) pool size is affected. Both NAD⁺ as well as NAD⁺/NADH regulate multiple critical reactions and cells strive to keep these constant to maintain cellular homeostasis. One hypothesis is that upon oxidation of NADH to NAD⁺ by mtLbNOX, NAD⁺ salvage is deactivated resulting in a stabilization of the NAD⁺/NADH redox ratio at the expense of NAD(H) pool size. The different effect of mtLbNOX compared to the original paper might be due to different cell types or due to a longer time after stable mtLbNOX overexpression, allowing for cellular allostasis.

Independent of the underlying reason why mtLbNOX modified pool size instead of NAD⁺/NADH ratio in our cells, here we just use it just as a tool to modulate NAD(H). Critically, in our cells, mtLbNOX has no effect on the NAD⁺/NADH ratio, thus allowing assessment of the effect of NAD(H) pool size on the mean NADH lifetime, further supporting our data with FK866.

To more accurately reflect this perspective, we have modified our manuscript (**page 5**, line 4) as follows:

“First, we biochemically assessed the effect of mtLbNOX expression on both NAD⁺ and NADH levels in 143B osteosarcoma cells. Contrary to Mootha et al.'s findings in their original paper characterizing mtLbNOX expressing cells, we found a significant decrease in both NAD⁺ and NADH levels (Fig.2c,d), possibly because the cell naturally equilibrates the redox state even when only NADH is scavenged by mtLbNOX. Although the underlying reason for this difference is still elusive, our mtLbNOX overexpressing cells can be used as an appropriate tool to study the effect of a reduced NAD(H) pool size but stable redox ratio on NADH FLIM.”

(4) In this study, the authors found that NR increased NAD⁺ and NADH levels, but the NAD⁺/NADH ratio may not be changed. But in Wang's paper, they found NAD⁺ precursor NR or NMN increased cytosolic NAD⁺/NADH ratio, not mitochondrial NAD⁺/NADH ratio (PMID: 34901920). So the author should describe it precisely, NR did not change mitochondrial or nucleus NAD⁺/NADH ratio, and talked about these results in the discussion.

We appreciate the insights into NR resulting in differential effects on the NAD⁺/NADH ratio in cytosol versus mitochondria. It is possible that cytosolic NAD⁺/NADH is increased upon NR in our cells but since the majority of NAD⁺/NADH is located in the mitochondria, the cytosolic change might not have been detected by our whole-cell assays. However, this might explain why NR had no significant effect on the nuclear NADH lifetime, since a redox-mediated increase in t_{mean} would counteract the increased pool-

size mediated decrease in τ_{mean} . Following the reviewer suggestions, we have included this possibility in the discussion. While we recognize the power of SoNar to elucidate redox changes on the subcellular level, we feel this experiment is not critical for the validity of our conclusion. In order to demonstrate that the shorter NADH lifetime upon NR treatment is mediated by a change in NAD(H) pool size, we need to exclude an effect of the NAD⁺/NADH ratio. Specifically, we need to exclude a more reduced NAD⁺/NADH ratio, which could induce a shorter τ_{mean} . We performed 3 independent approaches to quantify whole-cell NAD⁺/NADH and do not see any significant changes upon NR treatment, but if at all, a trend towards a higher NAD⁺/NADH ratio, which would rather induce a longer NADH lifetime. While this does not allow subcellular precision, it reassures that the NAD⁺/NADH ratio is not underlying the change in τ_{mean} upon NR.

We have modified our discussion (**page 10**, line 24) as follows to discuss our results in relation to Wang et al.'s findings:

“Another scenario when pool size changes might have masked changes in redox state is upon NR treatment. Wang et al. reported that NR causes an increase in cytosolic, but not mitochondrial NAD⁺/NADH ratio [citation]. We have observed a significant decrease in NADH lifetime in the mitochondria but not the nucleus/cytosol (Fig.1a). The more oxidized redox state in the cytosol would increase the NADH lifetime, while the increased NAD(H) pool would decrease the NADH lifetime, resulting in no effect of NR on the cytosolic τ_{mean} . In contrast, in the mitochondria the NAD⁺/NADH ratio does not change but the NAD(H) pool is increased, causing a shorter NADH lifetime. This highlights the strong interconnection of both redox state and NAD(H) pool size in altering the NADH fluorescence signal.”

(5) The authors did not describe details very clearly. For example, what is the concentration of NR or FK866 for NADH FLIM in cell? The glycolysis contributes NADH in cytosol. Why did the author focus on mitochondria and nucleus, not cytosol? How did the author use NADH FLIM to measure subcellular NADH? What are the targeted modifications?

We appreciate these valuable comments.

1. We have modified our methods section (**page 14**, line 40) to include NR and FK866 concentrations in the NADH FLIM section as follows:

“NR/FK866 treatments of varying concentrations (300 μ M and 1 mM NR; 1 nM, 1.5 nM, 2 nM, and 5 nM FK866) were administered in the same medium 24h prior to the experiments.”

2. In our experience, nuclear and cytoplasmic τ_{mean} are closely correlated as NAD⁺ and NADH move freely across pores in the nuclear envelope (White et al., 2012, PMID: 22436696).

To confirm this within our cells and experimental conditions, we reanalyzed FLIM images of untreated, NR treated, and FK866 treated cells for nuclear τ_{mean} and cytoplasmic τ_{mean} , confirming a close positive correlation between the values (graph below). This demonstrates that the nuclear NADH can be a good marker for both the nuclear and cytosolic NADH lifetime, while it is easier and more accurate to quantify compared to cytosolic regions in microscope images. To avoid confusion, we have included this graph as **Fig.S1** and added this justification in our manuscript as follows (**page 3**, line 20):

“The cytoplasmic NADH was not analyzed separately since it equilibrates with the nucleus through the nuclear pores [21]. This was confirmed by correlating nuclear and cytoplasmic mean NADH lifetimes across our treatment (Fig.S1).”

Figure S1. Cytoplasmic and nuclear τ_{mean} are closely correlated. *Cytoplasmic τ_{mean} correlated with nuclear in untreated, NR treated, and FK866 treated 143B cells. Each data point represents analysis from a single image with τ_{mean} averaged across 5 cells.*

3. We apologize for any confusion. Subcellular NADH was measured using a combination of thresholding and marking regions of interest in FLIM images. To more accurately describe this process, we've revised the methods section to the following (**page 15**, line 18):

“For subcellular analysis, nuclei were selected by drawing regions of interest (ROIs) using the “Define mask” tool in areas clearly distinguishable as nuclei by eye. At least five nuclei were masked for analysis per image. The mitochondria-rich regions were isolated from the nucleus and cytoplasm using the intensity threshold function because mitochondria tend to have higher fluorescence than the surrounding cytoplasm and the nucleus.”

4. We apologize for the confusion. We use the term *“targeted modifications”* to indicate that we treated cells to modify one factor (in this case NAD(H) pool size) as specifically as possible without impacting the redox state. We understand that our targeted treatments (mainly NR,

FK866, and mtLbNOX) inevitably have effects beyond the pool size modifications, however, by using the term “targeted modifications”, we intended to convey that those other effects are limited in comparison to the modifications in NAD(H) pool. We feel that “targeted” is a good word choice since it does not claim to be a highly “specific” modification, but we are open to a better phrasing.

Minor

(1) It is NAD⁺ not NAD⁺. It is mtLbNOX, not mtLBNOX.

Thank you for bringing our attention to these mistakes. We have corrected them in our manuscript accordingly (tracked in red).

Reviewer #2 (Remarks to the Author):

This work demonstrated the distinctive impacts of changes in NAD(H) redox state and NAD(H) pool size on FLIM parameters of NADH. The authors showed that both energy metabolism perturbations and changes in NAD(H) pool size impact the mean fluorescence lifetime of NADH, but they have different impacts on the short and long fluorescence lifetime and the free-to-bound NADH ratio, enabling the distinctive characterization of changes in energy metabolism and nucleotide pool size from non-invasive FLIM measurement of NADH. This work also cautions about the interpretation of NADH FLIM measurements given these two distinctive contributions. This is one of a rare study that systematically explores distinct contributions to FLIM of NADH, especially the effect of NAD(H) pool size that is mostly unaccounted for in the interpretation of FLIM measurements. Hence, I think this is an important contribution to the metabolic imaging field. To help clarify and improve the interpretation of the experiment, I have the following comments:

Major comments:

1. The authors segmented the NADH compartments into mitochondria and nucleus, but what about cytoplasm where glycolysis happens?

Thank you for this valuable comment. In our experience, nuclear and cytoplasmic τ_{mean} are closely correlated as NAD⁺ and NADH move freely across pores in the nuclear envelope (White et al., 2012, PMID: 22436696).

We reanalyzed FLIM images of untreated, NR treated, and FK866 treated cells for nuclear τ_{mean} and cytoplasmic τ_{mean} , confirming a positive correlation between the two values across the relevant conditions for this manuscript (graph below). Therefore, we elected to present the nuclear NADH

lifetime as marker for both nuclear and cytoplasmic NADH since the nuclear NADH is easier and more accurate to assess using regions of interests due to the larger size and more homogenous structure. To avoid confusion, we have included this graph as **Fig.S1** added this justification in our manuscript as follows (**page 3**, line 20):

“The cytoplasmic NADH was not analyzed separately since it equilibrates with the nucleus through the nuclear pores [21]. This was confirmed by correlating nuclear and cytoplasmic mean NADH lifetimes across our treatment (*Fig.S1*).”

Figure S1. Cytoplasmic and nuclear τ_{mean} are closely correlated. Cytoplasmic τ_{mean} correlated with nuclear in untreated, NR treated, and FK866 treated 143B cells. Each data point represents analysis from a single image with τ_{mean} averaged across 5 cells.

2. mtLbNOX is used to perturb NAD(H) pool size and is shown to leave NADH redox state unchanged. This is a bit counter intuitive to me because mtLbNOX is an NADH oxidase, and one would expect the exact opposite. Is there an argument for using mtLbNOX to perturb the NAD(H) pool size?

Thank you very much for this comment. We must admit that we were similarly surprised about this result, as we initially intended to employ mtLbNOX to alter the NAD^+/NADH ratio, not the pool size. However, as indicated above, multiple, independent methods to quantify NAD^+/NADH verified that in these cell types, mostly the NAD(H) pool size is affected. Both NAD^+ as well as NAD^+/NADH regulate multiple critical reactions and cells strive to keep these constant to maintain cellular homeostasis. One hypothesis is that upon oxidation of NADH to NAD^+ by mtLbNOX, NAD^+ salvage is deactivated resulting in a stabilization of the NAD^+/NADH redox ratio at the expense of NAD(H) pool size. The different effect of mtLbNOX compared to the original paper might be due to different cell types or due to a longer time after stable mtLbNOX overexpression, allowing for cellular allostasis.

Independent of the underlying reason why mtLbNOX modified pool size instead of NAD^+/NADH ratio in our cells, here we just use it just as a tool to modulate NAD(H). Critically, in our cells, mtLbNOX has no effect on the NAD^+/NADH ratio, thus allowing assessment of the effect of NAD(H) pool size on the mean NADH lifetime, further supporting our data with FK866.

To more accurately reflect this perspective, we have modified our manuscript as follows (page 5, line 4):

“First, we biochemically assessed the effect of mtLbNOX expression on both NAD⁺ and NADH levels in 143B osteosarcoma cells. Contrary to Mootha et al.’s findings in their original paper characterizing mtLbNOX expressing cells, we found a significant decrease in both NAD⁺ and NADH levels (Fig.2c,d), possibly because the cell naturally equilibrates the redox state even when only NADH is scavenged by mtLbNOX. Although the underlying reason for this difference is still elusive, our mtLbNOX overexpressing cells can be used as an appropriate tool to study the effect of a reduced NAD(H) pool size but stable redox ratio on NADH FLIM.”

3. NADH can also bind to mtLbNOX, which can potentially change its fluorescence lifetime. How to make sure the observed change of NADH fluorescence lifetime is not due to this potential confounding factor?

Thank you for this comment. We agree with the reviewer that binding of NADH to mtLbNOX may impact the fluorescence lifetime of NADH. Although we cannot exclude this possibility, when we correlate τ_2 , the protein-bound NADH lifetime, to mtLbNOX expression, there is no significant correlation. Therefore, we believe that mtLbNOX does not appreciably impact the protein-bound NAD(H) pool. This analysis is reflected in the graphs attached below, the second of which has been added as Fig.2e. Additionally, we have modified our manuscript as follows (page 5, line 35):

“Next, to rule out the possibility that binding of NADH to mtLbNOX was impacting the protein-bound NADH fluorescence lifetime, we plotted τ_2 , representative of the protein bound NADH lifetime, against the mtLbNOX expression (Fig.2e). Unlike τ_{mean} , τ_2 showed no significant correlation with mtLbNOX expression, suggesting that mtLbNOX expression does not significantly impact the protein-bound NADH lifetime.”

Figure 2. a) Mean NADH lifetime (τ_{mean}) in mitochondria of antimycin A treated 143B cells normalized to average τ_{mean} in all antimycin A treated cells (red), and in mitochondria of untreated 143B cells normalized to average τ_{mean} in all untreated cells (black), correlated to mtLbNOX expression **e)** Protein-bound NADH lifetime (τ_2) in mitochondria of antimycin A treated 143B cells normalized to average τ_2 in

all antimycin A treated cells (red), and in mitochondria of untreated 143B cells normalized to average τ_2 in all untreated cells (black), correlated to mtLbNOX expression.

4. The metabolic delta is a very interesting parameter that can potentially isolate the change of respiration from the change of NAD(H) pool size, but is mean fluorescence lifetime the best choice to compute this quantity? What about the free-to-bound NADH ratio, which is more sensitive to respiration perturbation and is potentially free of the artifacts suffered from the mean lifetime as explained in the manuscript?

We fully agree with the reviewer and thank them for this suggestion. As we have shown in Figure 5, free-to-bound NADH (a_1/a_2) is more altered upon respiration-induced lifetime changes and less upon pool-size induced lifetime changes. Thus, applying the idea of the metabolic delta on a_1/a_2 instead of τ_{mean} would provide a great approach to detect respiration. We will happily add this point into our discussion as follows (page 12, line 30):

“The differential changes in FLIM parameters with pool size vs respiration changes also inform additional methods for analyzing NADH FLIM data depending on the selected application. For example, when detection of respiration changes is desired, metabolic delta can be calculated using the ratio of free to protein-bound NADH (a_1/a_2) rather than τ_{mean} , as we show that a_1/a_2 is less sensitive to pool size alterations than τ_{mean} . Thus, the resulting values are less likely to be confounded by any pool size changes.”

We would like to express our gratitude to this reviewer for this valuable contribution and expert knowledge.

5. The mechanism how respiration perturbation and NAD(H) pool size perturbation impacts FLIM parameters distinctively remains elusive to me. How does NAD(H) pool size change short NADH lifetime more than respiration, given both perturbations could change pH or other microenvironment factors?

We thank reviewer 2 for this comment. At the moment, the mechanism of these differences are not fully elucidated, however, we feel that we have strong descriptive evidence for the validity of these observations. The differential changes in the FLIM parameters with respiration changes and with pool size changes are consistent not only across 2 different cell types (HEK293 and 143B), but also with different methods of altering NADH pool size (enzymatically via mtLbNOX and chemically via NR and FK866). We hypothesize that respiration changes and pool size changes may cause a differential distribution of NADH, thus altering the microenvironment. However, we recognize that these observations may be different in different cell types or in tissues *in-vivo*. More work in the future is

needed to elucidate whether FLIM parameters behave similarly across cell types and tissues upon specific modifications to respiration or NAD(H) pool size. Despite this, our work depicts an important stepping stone in first identifying the NAD(H) pool size as a factor that should be considered in NADH FLIM interpretations and providing a working hypothesis to differentiate the influencing factors.

To reflect this limitation, we have modified our results and discussion sections as follows:

Results (page 8, line 38):

“Although the exact mechanism behind these differential changes are not fully elucidated, the consistency of our observations across 2 cell lines and with both chemical (NR/FK866) and enzymatic (mtLbNOX) modifications of NAD(H) pool size, provides strong descriptive evidence for the validity of these observations.”

Discussion (page 12, line 20):

“Since the lifetime components τ_1 and τ_2 are sensitive to the microenvironment, such as viscosity and pH [13, 40], we hypothesize that changes in pool size result in a different subcellular distribution of NAD(H) compared to changes in respiration, causing differential changes in the cellular microenvironment. This results in specific changes in FLIM parameters, allowing differentiation of respiration-induced NADH lifetime alterations from pool size-induced alterations. Although the exact underlying molecular mechanisms behind these observations are not fully elucidated, they are consistent across 143B cells and HEK293 cells, as well as with 2 different methods of altering NAD(H) pool (chemically via NR/FK866 and enzymatically via mtLbNOX), providing strong descriptive evidence that the differential changes are genuine. However, it is possible that our findings may not be corroborated in different cell types or in tissues, thus, more work in the future is needed to determine whether FLIM parameters behave similarly across cell types and tissues upon targeted modifications to respiration or NAD(H) pool size. Despite this, our work depicts an important stepping stone in first identifying the NAD(H) pool size as a factor that should be considered in NADH FLIM interpretations and providing a working hypothesis to differentiate the influencing factors.”

Minor comments:

6. How to test that respiration perturbation impacts redistribution of NAD(H) across subcellular compartments?

We guess this comment refers to our finding that the nuclear NADH lifetime increases upon respiration perturbation. We specifically tried to phrase the NADH redistribution and reduced nuclear NAD(H) pool size as a possible interpretation for the otherwise inexplicable elongation of the nuclear NADH lifetime upon AA treatment. While this highly significant observation nicely fits our hypothesis, within the constraints of this paper we cannot ultimately show a causal relationship. Theoretically, one could inhibit the mitochondrial NAD⁺ transporter SLC25A51. This way, one can prevent NAD⁺ from being transported into and sequestered in the mitochondria. This should prevent the local decrease in the

nuclear/cytoplasmic NAD(H) pool and thus elongated NAD(H) lifetime upon antimycin A treatment. However, blocking SLC25A51 might have significant consequences on cellular metabolism and NADH distribution independent on antimycin A treatment, rendering this not feasible within this revision.

To avoid any false claims, we have ensured to phrase this section carefully.

7. In Figure 2a), AA does not seem to change the mean lifetime, is this as expected?

We apologize for the confusion in this figure. In Figure 2a, mean lifetime was normalized within each condition, so τ_{mean} for each untreated cell was normalized to the average τ_{mean} for all untreated cells, and τ_{mean} for each AA treated cell was normalized to average τ_{mean} for all AA treated cells. This was done as we were mainly interested in the trend of τ_{mean} with increased *mtLbNOX* expression, and not the absolute values. To help clarify this, we revised the figure caption to the following:

“**a)** Mean NADH lifetime (τ_{mean}) in mitochondria of antimycin A treated 143B cells normalized to average τ_{mean} in all antimycin A treated cells (red), and in mitochondria of untreated 143B cells normalized to average τ_{mean} in all untreated cells (black), correlated to *mtLbNOX* expression. **b)** Same as (a) for nuclear τ_{mean} . *mtLbNOX* expression was quantified by CayenneRFP fluorescence intensity.”

Below is the figure where τ_{mean} was not normalized in mitochondria, which shows the expected effect of antimycin A on τ_{mean} .

8. In the introduction, it is claimed that protein-bound NADH remains relatively stable, but Figure 5 and S6 seem to show significant change in this quantity?

We appreciate this comment and apologize for any confusion. Our intention was to express that protein-bound NADH remains relatively stable compared to free NADH. This is corroborated in Figure 5c

and Figure S8 (formerly S6), where the changes in a_{1abs} (free NADH) are much larger than the changes in a_{2abs} (protein-bound NADH) for both respiration and pool size induced changes. We recognized that the protein bound NADH pool does still change to some extent when cellular parameters are manipulated, and we have revised our introduction to reflect this as follows (page 2, line 21):

“Since the protein-bound NADH pool remains relatively stable *in comparison to the free NADH pool*, a larger protein-bound to free NADH ratio, or a longer mean NADH lifetime, indicates less NADH relative to NAD^+ and a more oxidized redox state.”

9. Following question 8, is the correlation between mean NADH lifetime and respiration activity always monotonic (after excluding the change of NAD(H) pool size)?

Thank you for this comment. The correlation between mean NADH lifetime and respiration is relatively monotonic after correction for other influencing factors (see our previous work on correlations between respiration and τ_{mean} (Schaefer et al., 2017)). However, the mean NADH lifetime is a weighted average between free NADH (typically 400 ps lifetime) and protein bound NADH (typically 2500 ps lifetime), so it must always be between 400 ps and 2500 ps. Thus, as τ_{mean} approaches either 2500 ps or 400 ps, similar changes in respiration do not produce equivalent changes in τ_{mean} . We observe this effect when τ_{mean} is altered as a result of pool size changes (Fig.S7a), and expect a similar result with respiration changes. However, respiration changes typically only induce changes in τ_{mean} within the range of 100-200 ps, so within this small range τ_{mean} can be expected to be relatively monotonic. To address this, we have included the following in our discussion (page 11, line 37)

“The same applies for the correlation of τ_{mean} with respiration. However, metabolic changes only induce 100-200 ps differences in τ_{mean} when factors such as pool size and pH are controlled for. Thus, the relationship between τ_{mean} and respiration is more likely to be monotonic in the relevant ranges [10].”

Reviewers' comments:

Reviewer #1 (Remarks to the Author):

The authors answered most of my questions, I have only 2 concerns.

(1) Although the authors used Blacker's method to determine NADPH/NADH ratio, they still did not separate the NADPH or NADH. If the authors did not plan to use indicators, I suggest the authors may isolate the cell compartments and then measure the subcellular NAD(P)(H) content by biochemical methods. Moreover, the authors should discuss the advantages of fluorescence lifetime imaging when compared to genetically encoded biosensors.

(2) It is still very confused that why mtLbNOX change the NAD(H) pool size. The authors may use another different method to verify NAD(H) levels again and discuss more about the potential reasons.

Reviewer #3 (Remarks to the Author):

The authors have satisfactorily addressed most of my questions, with three remaining comments:

1. It is good to show that nuclear tau_mean correlates with cytoplasmic tau_mean, but given the microenvironment is different for NADH in the cytoplasm and nucleus, as can be seen by the differences in the absolute values of tau_mean in Figure S1, I suggest the authors add the cytoplasmic data to Figure 1 a,b and Figure 2.

2. Regarding choosing free-to-bound NADH as an alternative metabolic delta, the authors can cite this related article: Yang et al. 2021, PMID: 34806591

3. Typos: "cytoplasmic" rather than "cytoplasmatic"

Reviewer #1 (Remarks to the Author):

The authors answered most of my questions, I have only 2 concerns.

(1) Although the authors used Blacker's method to determine NADPH/NADH ratio, they still did not separate the NADPH or NADH. If the authors did not plan to use indicators, I suggest the authors may isolate the cell compartments and then measure the subcellular NAD(P)(H) content by biochemical methods. Moreover, the authors should discuss the advantages of fluorescence lifetime imaging when compared to genetically encoded biosensors.

We highly appreciate the reviewer's suggestion to isolate cell compartments and check whether the treatments with NR and FK866 cause differential effects on the NADPH/NADH ratio in subcellular compartments. We isolated mitochondrial and cytosolic fractions from 143B cells. We decided to analyze the cytosol instead of the nuclei since we were unable to obtain a sufficiently large nuclear fraction free of unbroken cells and use the same samples to obtain mitochondrial fractions using existing protocols (DOI: 10.1186/1756-0500-5-513). We additionally collected whole-cell samples prior to subcellular fractionation as controls to confirm our previous results. The ratio of reduced nucleotides (NADH/NADPH) was quantified biochemically using NAD/NADH and NADP/NADPH Glo™ Assays from Promega and results were normalized to protein amount determined by BCA.

As expected, we found higher NADPH/NADH ratios in the cytosol compared to the mitochondria, which is in line with prior reports (DOI: 10.1074/jbc.TM117.000258). We also show that treatment with FK866 and NR has a similar effect on NADPH/NADH ratio in mitochondria and in cytosol compared to whole cells, indicating that neither NR or FK866 causes significantly differential changes to the NADPH/NADH ratio across the cellular compartments. Thus, their impact on subcellular NADH FLIM results is also minimal.

These results are shown below as **Fig.S5h** and our manuscript has been modified as follows (**page 4, line 25**):

“To verify our biochemical quantifications, we also performed analysis developed by Blacker et al. to determine the NADPH/NADH ratio in treated and untreated cells using lifetime components from FLIM [15] (Fig.S5d,e). This analysis showed close correlations between the optically quantified NADPH/NADH ratio in both mitochondria and the nucleus and biochemical quantifications of NADPH/NADH, suggesting no significant differential effects on the subcellular level (Fig.S5f,g). To confirm this, we isolated cytosolic and mitochondrial fractions from NR and FK866 treated cells, and biochemically quantified NADPH/NADH in each subcellular fraction and in whole cell samples collected prior to fractionation (Fig.S5h). In accordance with prior studies, NADPH/NADH is higher in cytosol than in mitochondria [22]. Crucially though, treatment with both FK866 and NR has a similar effect on NADPH/NADH in mitochondria and in cytosol compared to whole cells, reiterating that neither NR or FK866 causes differential changes to the NADPH/NADH ratio across cellular compartments. Thus, their impact on subcellular NADH FLIM results is also minimal.”

Additionally, to discuss the strengths of fluorescence lifetime imaging over genetically encoded biosensors, we have included the following addition to our discussion (page 10, line 19):

“Similarly, genetically-encoded biosensors, such as SoNar and iNap, also allow subcellular visualization of NAD(P)(H) [29,30]. The advantage of the biosensors is a high specificity for NADH over NADPH (or vice versa) as well as an easier calibration for comparison of absolute redox ratios across cell compartments/cell types. However, NADH FLIM does not require any additional cellular manipulations to express NAD(H) biomarkers, which may be challenging in sensitive or difficult to transfect cell lines. This also means that, unlike genetically encoded biosensors, FLIM can be used directly to quantify NAD(H) pool in tissues, as we have demonstrated in Fig.3, or even humans [31].”

Figure S5h NADPH/NADH ratio quantified biochemically upon 24 hour treatments of 5nM FK866 and 1mM NR in whole cells and in mitochondria and cytosol after subcellular fractionation. Significances between untreated and NR/FK866 treated samples are indicated with *, and significances between mitochondrial and cytosol within each treatment group (untr, NR, FK866) are indicated with #.

(2) It is still very confused that why mtLbNOX change the NAD(H) pool size. The authors may use another different method to verify NAD(H) levels again and discuss more about the potential reasons.

We repeated our quantification with a different biochemical kit (Sigma Aldrich NAD/NADH Quantification Kit) and a fresh aliquot of mtLbNOX expressing 143B cells. As seen below, both NAD⁺ and NADH levels are similarly decreased in comparison to control 143B cells (Fig.S6a,b) while the ratio of NADH to NAD⁺ is not altered (Fig.S6c), confirming the results of our previous quantification using the Promega NAD/NADH Glo™ Assay.

One hypothesis as to why our observations differ from Titov et al. (previously referenced as Mootha et al. according to last author), is that as mtLbNOX is stably expressed over a long period of time, the cells

are able to equilibrate NADH and NAD⁺ levels to rebalance redox ratio. To test this hypothesis, we quantified NAD⁺/NADH one week and two weeks (data in *Fig.S6a-c*) after starting selection pressure with G418 in our mt*LbNOX* cells (*Fig.S6d-f*). At the earlier 1-week time point, the redox ratio is more reduced in mt*LbNOX* expressing cells, which more closely matches Titov et al.'s results. However, unlike their results, NAD⁺ levels are also significantly decreased, suggesting that the cells had already begun to equilibrate their redox ratio. These results demonstrate that the time after transfection and the respective cellular response to *LbNOX* expression, which might differ from cell line to cell line, play a significant role in the effect of *LbNOX* on NAD(H) pool size and redox ratio.

We have included this data as **Figure S6** and modified our manuscript as follows (**page 5, line 16**):

“First, we biochemically assessed the effect of mt*LbNOX* expression on both NAD⁺ and NADH levels in 143B osteosarcoma cells. Contrary to Titov et al.'s findings in their original paper characterizing mt*LbNOX* expressing cells, we found a significant decrease in both NAD⁺ and NADH levels (*Fig.2d,e*). Subsequent testing with an alternate biochemical quantification kit confirmed these findings and demonstrated no significant difference in NAD⁺/NADH ratio between control and mt*LbNOX* expressing cells (*Fig.S6a-c*), possibly because the cell naturally equilibrates the redox state even when only NADH is scavenged by mt*LbNOX*. To test this hypothesis, we quantified NAD⁺ and NADH one week and two weeks (data in *Fig.S6a-c*) after starting selection pressure with G418 in our mt*LbNOX* cells (*Fig.S6d-f*). At the earlier 1-week time point, the redox ratio is more reduced in mt*LbNOX* expressing cells, which more closely matches Titov et al.'s results. However, NAD⁺ levels were already significantly decreased, suggesting that the cells had begun to equilibrate their redox ratio. These results demonstrate that the time after transfection and the respective cellular response to *LbNOX* expression play a significant role in the effect of *LbNOX* on NAD(H) pool size and redox ratio, and suggest that our mt*LbNOX* overexpressing cells can be used as an appropriate tool to study the effect of a reduced NAD(H) pool size but stable redox ratio on NADH FLIM.”

Figure S6. Redox ratio in mtLbNOX expressing cells equilibrates over time.

a-c) NAD⁺ levels (a), NADH levels (b), and NADH/NAD⁺ ratio (c) quantified biochemically with the NAD/NADH Quantification Kit (Sigma-Aldrich) in 143B control cells and 143B mtLbNOX expressing cells around 2 weeks after starting selection pressure. Data is pooled from 2 separate experiments. **d-f)** NAD⁺ levels (d), NADH levels (e), and NADH/NAD⁺ ratio (f) quantified biochemically 1 week and 2 weeks (data same as a-c) after starting selection pressure. The bars indicate the mean and standard error, and significances were calculated using t tests between selected groups and are indicated as n.s. for $p > 0.05$, ** for $p < 0.01$, and *** for $p < 0.001$.

Reviewer #2 (Remarks to the Author):

The authors have satisfactorily addressed most of my questions, with three remaining comments:

1. It is good to show that nuclear tau_mean correlates with cytoplasmic tau_mean, but given the microenvironment is different for NADH in the cytoplasm and nucleus, as can be seen by the differences in the absolute values of tau_mean in Figure S1, I suggest the authors add the cytoplasmic data to Figure 1 a,b and Figure 2.

We appreciate this suggestion and have accordingly added cytoplasmic data to **Figure 1a** and **1b** and in Figure 2 as **2c** as shown below. Small modifications in our manuscript to address these additions are tracked in red on **pages 3, 5, and 6**.

Figure 1a/b) Mean NADH lifetime (τ_{mean}) of mitochondria, nuclei, and cytoplasm in HEK293 cells upon 24h treatment with nicotinamide riboside (NR) (a) or FK866 (b) with exemplary images encoding τ_{mean} in false-colors (red = shorter, blue = longer).

Figure 2c) Mean NADH lifetime (τ_{mean}) in cytoplasm of antimycin A treated 143B cells expressing mtLbNOX normalized to average τ_{mean} in all antimycin A treated cells (red), and in cytoplasm of untreated 143B cells expressing mtLbNOX normalized to average τ_{mean} in all untreated cells (black), correlated to mtLbNOX expression. Representative FLIM images encode τ_{mean} in false-colors (red = shorter lifetime, blue = longer lifetime) and corresponding fluorescence intensity images show CayenneRFP fluorescence (used as a direct measure of mtLbNOX expression levels).

2. Regarding choosing free-to-bound NADH as an alternative metabolic delta, the authors can cite this related article: Yang et al. 2021, PMID: 34806591

Thank you for this suggestion, we have included it accordingly (page 13, line 18).

3. Typos: “cytoplasmic” rather than “cytoplasmatic”

Thank you for bringing our attention to this mistake! We have corrected it in our manuscript (page 3, line 26).

REVIEWERS' COMMENTS:

Reviewer #1 (Remarks to the Author):

The authors answered all my questions. I have no more concerns and suggest it publication.

Reviewer #3 (Remarks to the Author):

The authors have satisfactorily addressed all my questions. The paper is very rigorous.